# STYLE-COHERENT MULTI-MODALITY IMAGE FUSION

## ABSTRACT

Multi-modality image fusion (MMIF) integrates heterogeneous images from diverse sensors. However, existing MMIF methods often overlook significant style discrepancies, such as saturation and resolution differences between modalities, resulting in overly smooth features in certain modalities. This tendency causes models to misjudge and disregard potentially crucial content. To address this issue, this paper proposes a novel style-coherent multi-modality fusion model that adeptly merges heterogeneous styled features from various modalities. Specifically, the proposed style-normalized fusion module progressively supplements the complete content structure by merging style-normalized features during cross-modal feature extraction. Meanwhile, a style-alignment fusion module is developed to align different feature representations across modalities, ensuring consistency. Additionally, to better preserve information and emphasize critical patterns during fusion, an adaptive reconstruction loss is applied to multi-modal images transformed into a unified image domain, enforcing mapping to a consistent modality representation. Extensive experiments validate that our method outperforms existing approaches on multiple MMIF tasks and exhibits greater potential to facilitate downstream applications.

## 1 INTRODUCTION

Multi-modality image fusion (MMIF) aims to generate visually enhanced images by integrating complementary details from multi-modal images of the same scene captured by different devices (Liu et al., 2020; He et al., 2023a; Yan et al., 2022; Zhou et al., 2023b). The versatility of MMIF, arises from the diversity of imaging sensors, making it applicable to a wide range of tasks such as infrared and visible image fusion (IVF), medical image fusion (MIF), and biological image fusion (BIF). These fused images offer more discernible representations of objects and scenes, benefiting applications like visual enhancement (Liu et al., 2020; He et al., 2023a; Yan et al., 2022; Zhou et al., 2023b), image registration (Jiang et al., 2022; Wang et al., 2022; Xu et al., 2022), and object semantic segmentation (Zhou et al., 2022; Liu et al., 2023b; 2024b; Zhang et al., 2024).

Multi-modal images have distinct visual characteristics like differences in saturation, resolution, and spectral properties due to variations in external styles. For instance, in IVF, visible images are sensitive to illumination changes, while infrared images are generally characterized by noise and lower resolution. To address the limitations of individual source images and produce a comprehensive composite, MMIF should effectively balance these external styles while preserving the unified internal content from all contributing sources. Recently, the rapid development of deep learning has led to various learning-based MMIF methods (Jung et al., 2020; Xu et al., 2020b;a; Zhang et al., 2020a;b) have been proposed and demonstrated promising results (Li & Wu, 2018; Li et al., 2021; Zhao et al., 2020; Liang et al., 2022; Zhao et al., 2023b; Liu et al., 2023a). However, existing MMIF methods tend to overlook the significant style discrepancies like saturation and resolution between modalities and produce oversmooth features for a certain modality. Such a tendency causes models to misjudge and mistakenly disregard potential salient contents, which affects fusion quality and could compromise safety for night driving or medical uses. As shown in Fig. 1 (a), it leads to coarse intra-modal feature representations and significant disparities between cross-modal features, resulting in existing models producing fused features and results with diminished details and textures, specifically in elements like the stone pillars along the street and the cyclist with a taillight.

To mitigate the above issues, we propose a Style-coherent Content Fusion model (SCFNet). This approach transforms heterogeneous features from multiple modalities into a shared style-coherent

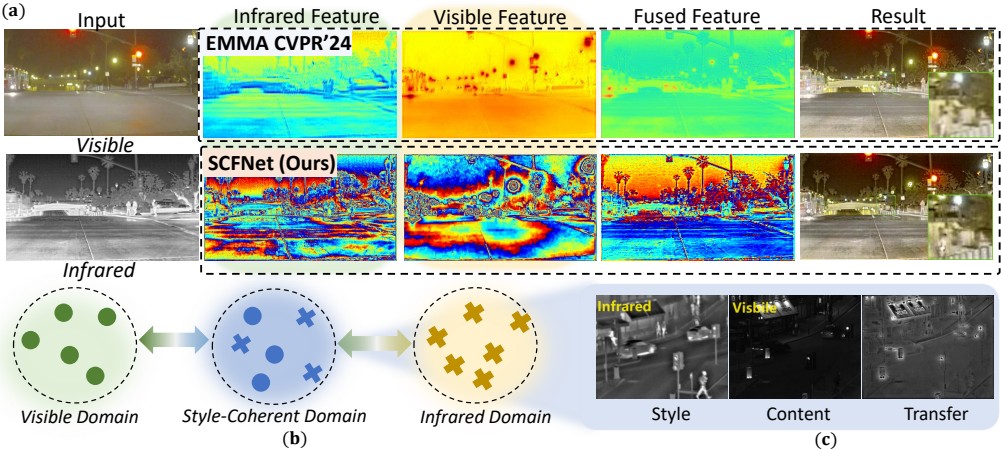

Figure 1: (a) Existing methods overlook style variations in MMIF, resulting in coarse and distinctly different multi-modal features that lose detail in fusion. (b) Our main idea: style-coherent content fusion model (SCFNet) integrates heterogeneous multi-modal features in the style-coherent latent domain by unifying variations styles. (c) Cross-modal style transfer: The transformation of visual characteristics can be accomplished using feature amplitude transfer as Lee et al. (2023).

latent domain, facilitating comprehensive content fusion as shown in Fig. 1 (b). We present evidence supporting this foundation (please refer to Fig. 1 (c)), motivated by observations that style and content in cross-domain features are inherently linked to frequency domain components (Lee et al., 2023). Specifically, we first develop a Style-Normalized Fusion (SNF) module that enhances content information through an effective style-content separation during the feature extraction process. By integrating the phases of style-normalized features, SNF progressively mitigates stylistic discrepancies and complements the invariant contents across modalities. Furthermore, to better align fused features and selectively focus on critical information, which is especially vital in fields like medical diagnosis where detail and texture are crucial, we introduce a Style-Alignment Fusion (SAF) module. This module aligns style-variant latent representations into a well-defined, modality-specific feature space. This aligned and regularized fused feature space retains more detailed information while preserving essential image priors.

On the other hand, the lack of ground truth (GT) makes it challenging to ensure the quality and reliability of the fused images. Some methods guide MMIF models based on priors from downstream tasks (Liu et al., 2022; 2023b; Sun et al., 2022; Tang et al., 2022a; Zhao et al., 2023a; Wang et al., 2022; Xu et al., 2020a; 2023b; Huang et al., 2022; Liu et al., 2024b; Zhang et al., 2024) or image priors from the pre-trained models (Zhao et al., 2023c; Yi et al., 2024). These approaches rely on explicit external guidance, which limits their generalizability. Therefore, this paper focuses on developing a self-supervised loss function for MMIF, requiring only source images collected in a general sensing setup. Some methods utilize crafted loss functions *e.g.,* the pixel-based loss (Zhao et al., 2023b; 2021; Liang et al., 2022) and gradient-based loss (He et al., 2023b; Xu et al., 2023a), applied directly to source images to maintain visible and texture fidelity. Due to inherent differences in source images, such as intensity and contrast, directly blending them as a supervision signal leads to smoother model outputs that lose important details. Therefore, we propose an adaptive reconstruction loss function that utilizes a learnable rescaling transformation for multi-modal images, allowing for comprehensive content supervision within a specific source domain. This linear transformation minimizes potential damage to the prior image and better unifies the diverse representations of scene information integrity. In addition, it encourages the model to learn identity mapping within this specific image domain, further promotes the retention of critical details.

Overall, the main contributions of this paper are summarized as follows:

• We propose a style-coherent content fusion model (SCFNet) that generates fused images with complete scene information and consistent external characteristics by unifying the external styles of heterogeneous modalities. To the best of our knowledge, this is the first work of MMIF grounded in a style-based foundation.

- To enhance and consolidate invariant content representations in a style-coherent latent space, we design the style-normalized fusion (SNF) module and the style-alignment fusion (SAF) module. To produce domain-specific supervisory signals that preserve content integrity, we propose an adaptive reconstruction loss.
- Our method attains state-of-the-art performance across various datasets. Furthermore, we substantiate the efficacy of SCFNet in multiple fusion tasks and support downstream applications.

## 2 RELATED WORK

**Multi-Modality Image Fusion.** Existing deep-learning based MMIF methods are roughly divided into two categories: deep unrolling networks (DUN) and deep neural networks (DNN). DUNs address the ill-posedness of MMIF by unfolding iterative algorithms into neural network layers (Deng & Dragotti, 2020; Gao et al., 2022; Xu et al., 2023a; Zhao et al., 2021; Ju et al., 2022; He et al., 2023b). However, it is challenging to model the complex non-linear degradation process of MMIF. DNNs learn nonlinear mappings to fuse diverse modalities (Jung et al., 2020; Xu et al., 2020b;a; Zhang et al., 2020a;b; Liu et al., 2024a). Generative adversarial network-based methods (Goodfellow et al., 2014; Liu et al., 2022; Ma et al., 2020; 2019; Zhang et al., 2021) constrain the distribution of the fused image to be similar to the input images for perceptually satisfactory. Encoder-decoder-based methods (Li & Wu, 2018; Li et al., 2021; Zhao et al., 2020; Liang et al., 2022; Zhao et al., 2023b; Liu et al., 2023a), employing CNN/Transformer blocks, learn feature fusion by translating between image and latent spaces. These methods overlook cross-domain stylistic discrepancies, directly proceeding with fusion and thereby compromising the integrity of the content.

To address the absence of GT, some methods guide the fusion process through image registration (Jiang et al., 2022; Wang et al., 2022; Xu et al., 2020b; 2023a; Huang et al., 2022) or downstream tasks such as object detection and semantic segmentation (Liu et al., 2022; 2023b; Sun et al., 2022; Tang et al., 2022a; Zhao et al., 2023a; Xu et al., 2020a; 2023b; Liu et al., 2024b). Some methods utilize perceptual priors (Goodfellow et al., 2014; Liu et al., 2022; Ma et al., 2020; 2019; Zhang et al., 2021) or natural image statistics (Zhao et al., 2023c; Yi et al., 2024) which are introduced by pre-trained models to obtain perceptually and visually satisfying images. Recently, DeRUN (He et al., 2023b) introduces a gradient direction-based entropy loss that effectively captures and represents textural details within images by focusing on the directional patterns of gradients. EMMA (Zhao et al., 2024) introduce an equivariant image loss that utilizes an image equivariance prior to constrain the structural properties of fused images. These aforementioned self-supervised methods do not explicitly address inconsistent supervised signal distributions caused by inherent differences.

**Disentanglement for Multi-Modal Tasks.** For modalities with distinct physical carriers like audio, text and images, existing methods Yao et al. (2024); Fei et al. (2021); Lee & Pavlovic (2021); Chen et al. (2023); Ouyang et al. (2021) disentangle the representations of features both within and across the different modalities. The goal is to obtain robust and independent feature representations that enhance the relevance of the outcomes for objectives. For MMIF, Some methods Zhao et al. (2023b; 2020); Liang et al. (2022); Xu et al. (2021) decouple sources into common and unique features to enhance feature integration. CDD Zhao et al. (2023b) employs shared and specific multi-modal feature decomposition, followed by separate fusions of specific and shared features. DIDFuse Zhao et al. (2020) uses four distinct encoders to extract scenario features and attribute latent representations, and then fuse each set. However, these MMIF methods overlook the unification of appearance styles across modalities and do not explicitly ensure the integrity of the content during fusion.

**Style-Based Learning.** Style-based learning methods consider domain differences as divergent stylistic attributes across domains (Huang & Belongie, 2017; Nam & Kim, 2018; Pan et al., 2018). These methods aim to extract features that exhibit invariance to stylistic variations, thereby enabling effective style transfer across different scenes. Based on frequency analyses, the style and content of an image are represented by the amplitude and phase in the frequency domain (Hansen & Hess, 2007; Oppenheim & Lim, 1981; Piotrowski & Campbell, 1982; Zhou et al., 2023a; Li et al., 2024). Recently, it has been demonstrated that the properties of amplitude and phase extend to feature representations (Lee et al., 2023). The style and content components (Lee et al., 2023) enable the adjustment of style and content magnitudes by modulating the amplitude and phase of the pre-normalized features, facilitating the learning of generalized features. In contrast, this paper devotes

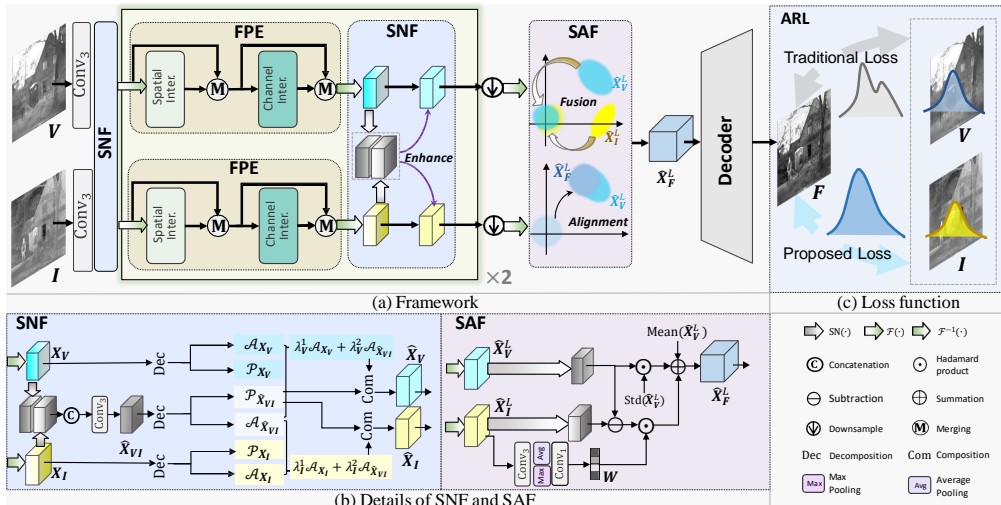

Figure 2: The overall architecture of our style-coherent content fusion model. (a) The framework mainly consists of three modules:Fourier Prior Embedding (FPE) model, style-normalized fusion (SNF) module and style-alignment fusion (SAF) module. (b) FPE captures the frequency information of features, SNF enhances and completes content information across multiple levels between encoders by normalizing styles, and SAF fuses the features extracted by the dual-branch encoder within a style-specific space. (c) The proposed adaptive reconstruction loss (ARL) function avoids ambiguity caused by supervising with inherently different source multi-modality images.

to exploring the potential effect of style learning theory in the field of MMIF by regarding the heterogeneity challenge of cross-modal fusion as the problem of style-consistent content integration.

## 3 METHODOLOGY

### 3.1 OVERVIEW

As illustrated in Fig. 2(a), our proposed SCFNet adopts a framework with a dual-branch encoder and a decoder. Considering the challenges that existing CNN and transformer blocks (Zhao et al., 2023b; 2020; Liang et al., 2022) encounter in extracting essential frequency features due to disruptions in local feature and global dependency coherence across modalities, we utilize the Fourier Prior Embedding (FPE) block (Zhou et al., 2023a). This block can efficiently capture the frequency information of features, thereby facilitating frequency-based style-content decomposition within our core module. See *Suppmentary* for details of FPE. Overall, the encoder has three main components: Fourier prior embedding block, style-normalized fusion (SNF) module, and style-alignment fusion (SAF) module. To reduce the complexity and potential discrepancies that arise from separate configurations for each modality, the twin encoder branches share the same structure and parameters. The decoder utilizes the standard architecture as Zhao et al. (2023b); Zamir et al. (2022). We take the IVF task as an example. Given a paired visible image $V$ and an infrared image $I$, the fusion process to obtain the fused image $F$ is described by $F = f(V, I)$, where $f(\cdot)$ represents the proposed SCF.

### 3.2 STYLE-NORMALIZED FUSION MODULE

SNF fuses the complete intrinsic structures of heterogeneous features through style normalization, enhancing the content information of modal features from the dual branches, as shown in Fig. 2. Based on the theoretical foundation of frequency domain style-learning (Lee et al., 2023), style-normalized representations are first achieved by substituting the amplitude, which represents the style, of features with that derived from normalized features.

To elaborate, consider a feature $X \in \mathbb{R}^{C \times H \times W}$, the fast Fourier transform (FFT) of the feature $X$ denoted as $\mathcal{F}(X)$ and the inverse FFT denoted as $\mathcal{F}^{-1}(X)$, applied to each channel independently. We denote the process of obtaining the amplitude $\mathcal{A}$ and phase $\mathcal{P}$ from the feature frequency as

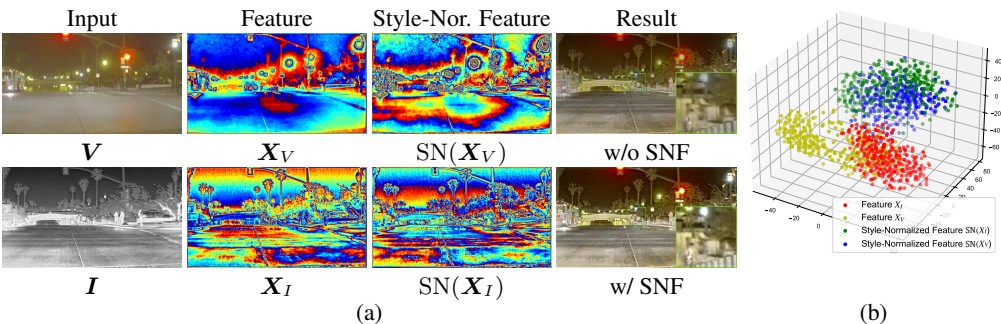

Figure 3: (a) From left to right: multi-modal images, original features, style-normalized features and fusion results. The low-contrast and low-intensity visible light features result in the smoothing of the fused image. (b) The t-SNE result of features. The style-normalized multi-modal features become closer than the original pair in terms of visualization and distribution.

decomposition operator $\text{Dec}(\cdot)$ with its inverse as composition operator $\text{Com}(\cdot)$:

$$[\mathcal{A}_{\boldsymbol{X}}, \mathcal{P}_{\boldsymbol{X}}] = \text{Dec}(\boldsymbol{X}), \quad \boldsymbol{X} = \text{Com}(\mathcal{A}_{\boldsymbol{X}}, \mathcal{P}_{\boldsymbol{X}}). \tag{1}$$

The normalized feature $\text{N}(\boldsymbol{X}) \in \mathbb{R}^{C \times H \times W}$ is obtained via instance normalization as:

$$\text{N}(\boldsymbol{X}) = \big(\boldsymbol{X} - \text{Mean}(\boldsymbol{X})\big)/\text{Std}(\boldsymbol{X}), \tag{2}$$

where $\text{Mean}(\cdot)$ and $\text{Std}(\cdot)$ are the mean and standard deviation functions along the channel dimension. This normalization impacts the phase component through the mean shift, which results in the deficiency of content information (Lee et al., 2023). To obtain a content-invariant representation, the style-normalized feature is obtained by composing the amplitude of $\text{N}(\boldsymbol{X})$ and the invariant phase of $\boldsymbol{X}$, as follows:

$$\text{SN}(\boldsymbol{X}) = \text{Com}(\mathcal{A}_{\text{N}(\boldsymbol{X})}, \mathcal{P}_{\boldsymbol{X}}). \tag{3}$$

Such style normalization maps different style features into a shared space, reducing discrepancies between modalities as illustrated in Fig 3. This facilitates the easier consolidation of complementary details from the content representations.

Then, SNF facilitates feature communication and enhancement between the twin encoders by fusing the content information of their style-normalized features. Concretely, the dual-branch encoder extracts hierarchical features at different levels from inputs $\boldsymbol{V}$ and $\boldsymbol{I}$. For simplicity, we denote the features from the visible and infrared streams as $\boldsymbol{X_V} \in \mathbb{R}^{C \times H \times W}$ and $\boldsymbol{X_I} \in \mathbb{R}^{C \times H \times W}$. We obtain their style-normalized representations $\text{SN}(\boldsymbol{X_V})$ and $\text{SN}(\boldsymbol{X_I})$ through Eq. 3. Within a shared style-normalized space, these style-normalized features are aggregated to obtain an enriched content embedding $\hat{\boldsymbol{X}}_{\boldsymbol{VI}} \in \mathbb{R}^{C \times H \times W}$ through:

$$\hat{\boldsymbol{X}}_{\boldsymbol{VI}} = \text{Conv}_3\Big(\text{Cat}\big(\text{SN}(\boldsymbol{X_V}), \text{SN}(\boldsymbol{X_I})\big)\Big), \tag{4}$$

where $\text{Cat}(\cdot)$ denotes the channel concatenation and $\text{Conv}_3$ is a $3 \times 3$ convolution layer.

To enrich the content integrity of representations from both encoders, we replace content by substituting the phase components of the original features with the phase of the content-complete fused style-normalized feature $\hat{\boldsymbol{X}}_{\boldsymbol{VI}}$. The degree of style modification is gradually adjusted by introducing learnable parameters $(\lambda_{\boldsymbol{V}}^1, \lambda_{\boldsymbol{V}}^2)$ and $(\lambda_{\boldsymbol{I}}^1, \lambda_{\boldsymbol{I}}^2)$. The enhanced features are obtained through:

$$\hat{\boldsymbol{X}}_{\boldsymbol{V}} = \text{Com}\big(\lambda_{\boldsymbol{V}}^1 \mathcal{A}_{\boldsymbol{X_V}} + \lambda_{\boldsymbol{V}}^2 \mathcal{A}_{\hat{\boldsymbol{X}}_{\boldsymbol{VI}}}, \mathcal{P}_{\hat{\boldsymbol{X}}_{\boldsymbol{VI}}}\big), \tag{5}$$

$$\hat{\boldsymbol{X}}_{\boldsymbol{I}} = \text{Com}\big(\lambda_{\boldsymbol{I}}^1 \mathcal{A}_{\boldsymbol{X_I}} + \lambda_{\boldsymbol{I}}^2 \mathcal{A}_{\hat{\boldsymbol{X}}_{\boldsymbol{VI}}}, \mathcal{P}_{\hat{\boldsymbol{X}}_{\boldsymbol{VI}}}\big), \tag{6}$$

where learnable parameters $\lambda_{\boldsymbol{V}}^1, \lambda_{\boldsymbol{V}}^2, \lambda_{\boldsymbol{I}}^1, \lambda_{\boldsymbol{I}}^2 \in \mathbb{R}^{C \times 1 \times 1}$ modulate between the normalized and original styles for both branches. By dynamically adjusting the feature style, SNF balances preserving the characteristics of source modal features with unifying cross-modal feature properties. This not only avoids deficiencies in feature content information caused by specific characteristics during the feature extraction process but also narrows feature discrepancies. Overall, SNF, with incremental feature enhancement, facilitates the gradual enrichment of content information throughout the hierarchical extraction process.

### 3.3 STYLE-ALIGNMENT FUSION MODULE

SAF prioritizes the alignment of the source modality image domain that contains more detailed and crucial information for recognition and understanding. In IVF, we aim to fuse multi-modal features aligning with the visible domain. This alignment is achieved by projecting the features $\hat{\boldsymbol{X}}_V^L$ and $\hat{\boldsymbol{X}}_I^L$ from the encoders at the last level $L$ into a shared embedding space with a normalized style as defined in Eq. 3. Within this space, the features are fused in a channel-weighted manner using $W$ to maintain spatial consistency. The alignment of the fused feature with the visible domain is then performed by adjusting based on the fundamental properties of the distribution, ensuring that the mean matches $\text{Mean}(\hat{\boldsymbol{X}}_V^L)$. $\text{Std}(\hat{\boldsymbol{X}}_V^L)$ aids $W$ in adjusting the standard deviation.

Specifically, the infrared feature $\hat{\boldsymbol{X}}_I^L$ is spatially squeezed through two types of pooling techniques *i.e.,* max pooling and average pooling. The squeezed features are then passed through a convolutional layer to generate channel-wise weights $\boldsymbol{W} \in \mathbb{R}^{C \times 1 \times 1}$, which can be expressed as:

$$\boldsymbol{W} = \text{Conv}_1\Big(\text{Cat}\big(\text{MaxPool}(\hat{\boldsymbol{X}}_I^L), \text{AvgPool}(\hat{\boldsymbol{X}}_I^L)\big)\Big), \tag{7}$$

where MaxPool, AvgPool are max pooling, average pooling, respectively. $\text{Conv}_1$ is a $1 \times 1$ convolution layer. Alignment of the cross-domain fused representation $\hat{\boldsymbol{X}}_F^L$ is achieved by adjusting its distributional statistics to match those of the target domain. The process is formulated as:

$$\hat{\boldsymbol{X}}_F^L = \boldsymbol{W} \odot \big(\text{SN}(\hat{\boldsymbol{X}}_I^L) - \text{SN}(\hat{\boldsymbol{X}}_V^L)\big) + \text{Std}(\hat{\boldsymbol{X}}_V^L) \odot \text{SN}(\hat{\boldsymbol{X}}_V^L) + \text{Mean}(\hat{\boldsymbol{X}}_V^L), \tag{8}$$

where $\odot$ denotes the dot product, $\text{Mean}(\hat{\boldsymbol{X}}_V^L) \in \mathbb{R}^{C \times 1 \times 1}$ is the channel-wise mean, and $\text{Std}(\hat{\boldsymbol{X}}_V^L) \in \mathbb{R}^{C \times 1 \times 1}$ is the channel-wise standard deviation. This facilitates effective integration and spatial correspondence, allowing comprehensive content transfer from the infrared to the visible domain.

### 3.4 ADAPTIVE RECONSTRUCTION LOSS FUNCTION

Directly self-supervising with source images, which have significant differences, leads to ambiguous solutions. Therefore, the proposed adaptive reconstruction loss function (ARL) employs the linearly rescaled source images as the supervision signal to guide the model in maintaining both stylistic coherence and content fidelity in the fused outputs. The learnable rescale function $\mathcal{R}(\cdot)$ is designed to adjust the distribution statistics of the source images, as follows:

$$\mathcal{R}(\boldsymbol{X}) = \alpha(\boldsymbol{X})\text{N}(\boldsymbol{X}) + \beta, \tag{9}$$

where $\alpha(\boldsymbol{I}) \in \mathbb{R}^+$ is the learnable parameter to scale and $\beta \in \mathbb{R}^+$ is the parameter to shift.

In conjunction with SAF, input source images are regularized as supervision signals in the visible image domain. The parameter $\beta$ is set to $\text{Mean}(\boldsymbol{V})$, anchoring the fused output to the distribution of the visible image domain. Meanwhile, the learnability of $\alpha$ provides significant flexibility, enabling dynamic adaptation to the distributional differences between the multi-modal inputs. It is obtained in a data-driven manner based on pooling and convolutional operations, as follows:

$$\alpha(\boldsymbol{X}) = \text{Conv}_1\Big(\text{Cat}\big(\text{MaxPool}(\boldsymbol{X}), \text{AvgPool}(\boldsymbol{X})\big)\Big). \tag{10}$$

To ensure appropriate information entropy and maintain balance, we make $\alpha(\boldsymbol{X})$ fall into $[\text{Max}\big(\text{Std}(\boldsymbol{V}), \text{Std}(\boldsymbol{I})\big), \text{Std}(\boldsymbol{X})]$ using the clip operator. The lower bound prevents the loss of essential details by avoiding overly narrow distribution, while the upper bound prevents the introduction of unrealistic artifacts due to excessively broad variability. Overall, this entropy-aware clipping provides controlled supervision while preserving informational integrity. This entropy-aware clipping thus provides controlled supervision while preserving the informational integrity of the fused output.

To further align with the characteristics of the visible image, the model is supervised within the visible image domain through identity mapping. By minimizing $\mathcal{L}_{AR}$, the model actively promotes the alignment of the prediction image with the source images, both characteristically and structurally, thus ensuring a high-fidelity reproduction of the characteristics of the visible domain in the fused output. The total loss contains a balancing factor $\gamma$ to weigh fusion and mapping losses, and $\rho$ to balance intensity and gradient terms as (Zhao et al., 2023b), defined as:

$$\begin{aligned} \mathcal{L}_{AR} = \|\boldsymbol{f}(\boldsymbol{V}, \boldsymbol{I}) - \text{Max}(\mathcal{R}(\boldsymbol{V}), \mathcal{R}(\boldsymbol{I}))\|_1 + \rho \left\|\|\nabla \boldsymbol{f}(\boldsymbol{V}, \boldsymbol{I})| - \text{Max}\big(|\nabla \mathcal{R}(\boldsymbol{V})|, |\nabla \mathcal{R}(\boldsymbol{I})|\big)\right\|_1 \\ + \gamma\big(\|\boldsymbol{f}(\boldsymbol{V}, \boldsymbol{V}) - \boldsymbol{V}\|_1 + \rho \||\nabla \boldsymbol{f}(\boldsymbol{V}, \boldsymbol{V})| - |\nabla \mathcal{R}(\boldsymbol{V})|\|_1 \big). \end{aligned} \tag{11}$$

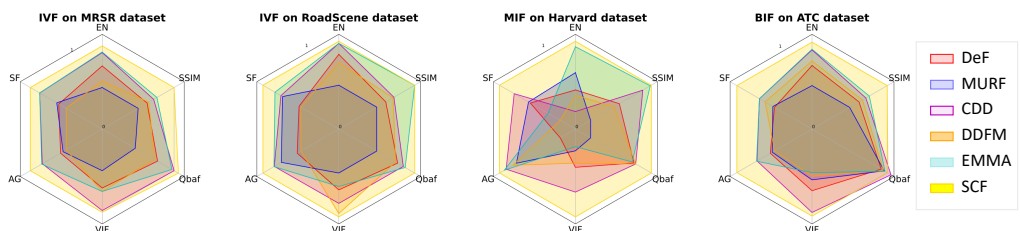

Figure 4: Quantity Comparison of IVF, MIF and BIF.

Table 1: Quantitative results of IVF on MSRS and RoadScence datasets.

| Methods | IVF on MSRS Dataset | | | | | | | IVF on RoadScene Dataset | | | | | | |
| | EN↑ | SD↑ | SF↑ | AG↑ | Qbaf↑ | VIF↑ | SSIM↑ | EN↑ | SD↑ | SF↑ | AG↑ | Qbaf↑ | VIF↑ | SSIM↑ |
|---|---|---|---|---|---|---|---|---|---|---|---|---|---|---|
| TarD | 5.28 | 25.22 | 5.98 | 1.83 | 0.41 | 0.42 | 0.45 | 7.26 | 47.44 | 11.11 | 4.14 | 0.40 | 0.56 | 0.88 |
| DeF | 6.46 | 37.63 | 8.60 | 2.80 | 0.56 | 0.77 | 0.92 | 7.36 | 47.03 | 10.99 | 4.38 | 0.48 | 0.63 | 0.89 |
| MURF | 6.07 | 26.82 | 8.91 | 2.67 | 0.46 | 0.46 | 0.81 | 6.91 | 33.34 | 13.88 | 5.37 | 0.43 | 0.52 | 0.79 |
| MetaF | 5.65 | 24.97 | 9.99 | 3.40 | 0.48 | 0.47 | 0.78 | 6.88 | 31.97 | 13.38 | 5.57 | 0.35 | 0.58 | 0.80 |
| CDD | 6.70 | 43.38 | _11.56_ | 3.73 | _0.69_ | _1.05_ | 1.00 | _7.52_ | 54.42 | 14.17 | 5.81 | _0.52_ | _0.66_ | _0.98_ |
| DDFM | 6.19 | 29.26 | 7.44 | 2.51 | 0.58 | 0.73 | 0.94 | 7.24 | 42.43 | 10.68 | 4.15 | 0.55 | 0.62 | 0.97 |
| SegM | 5.95 | 37.28 | 11.10 | 3.47 | 0.64 | 0.88 | 0.95 | 7.29 | 46.14 | 14.47 | 5.57 | 0.52 | 0.65 | 0.97 |
| EMMA | _6.71_ | 44.13 | _11.56_ | _3.76_ | 0.58 | 0.97 | _1.04_ | _7.52_ | 54.81 | 15.21 | _5.83_ | 0.47 | _0.66_ | **1.21** |
| **SCFNet** | **6.82** | **52.34** | **13.01** | **4.34** | **0.70** | **1.06** | **1.25** | **7.55** | **55.29** | **17.32** | **6.52** | **0.56** | **0.72** | **1.21** |

## 4 EXPERIMENTS

We first describe the implementation detail of SCFNet. Then we compare our proposed SCFNet with the state-of-the-art (SOTA) methods across different datasets of tasks. Fig 4 illustrates the superior performance of our SCFNet method across almost all metrics compared to other MMIF methods on IVF, MIF and BIF. Additionally, we validate our contributions in downstream tasks. To identify the contribution of each component, we further perform more analysis and ablation study.

### 4.1 EXPERIMENTAL SETTINGS

**Implementation Details.** The SCFNet is implemented with a batch size of $8$. Adam optimization is called for 200 epochs with an initial learning rate of $10^{-4}$, decayed by $0.9$ every 20 epochs. As for loss functions Eq. 11, the loss weight $\gamma$ is set to $0.1$ and $\rho$ is set to $2$.

**Compared Methods and Metrics.** We compare SCFNet with eight SOTA methods, including TarD (Liu et al., 2022), DeF (Liang et al., 2022), MURF (Xu et al., 2023b), MetaF (Zhao et al., 2023a), CDD (Zhao et al., 2023b), DDFM (Zhao et al., 2023c), SegM (Liu et al., 2023b) and EMMA (Zhao et al., 2024). Fusion results are quantitatively evaluated using seven metrics, including entropy (EN), standard deviation (SD), spatial frequency (SF), visual information fidelity (VIF), edge-based similarity measurement (Qbaf), average gradient (AG), and structural similarity index (SSIM). The best in tables is in **boldfaced**, and the second-best is underlined.

### 4.2 INFRARED AND VISIBLE IMAGE FUSION

Following (Zhao et al., 2023b; 2024; Tang et al., 2022b), we utilize 1083 image pairs from MSRS (Tang et al., 2022b) dataset for training, and 50 image pairs from RoadScene (Xu et al., 2020b) dataset for validation. The model is trained with patches of size $128 \times 128$. To assess model generalization, we evaluate performance across three diverse datasets, consisting of 361 image pairs from the MSRS (Tang et al., 2022b) dataset, 50 image pairs from the RoadScene (Xu et al., 2020b) dataset, and 25 image pairs from the TNO (Toet & Hogervorst, 2012) dataset.

The quantitative results are listed in Tab. 1 and Tab 2, which are quoted from (Zhao et al., 2024) or obtained with released codes. Our SCFNet demonstrates consistently superior performance over existing MMIF methods across most evaluated metrics on three distinct datasets. This consistent out-

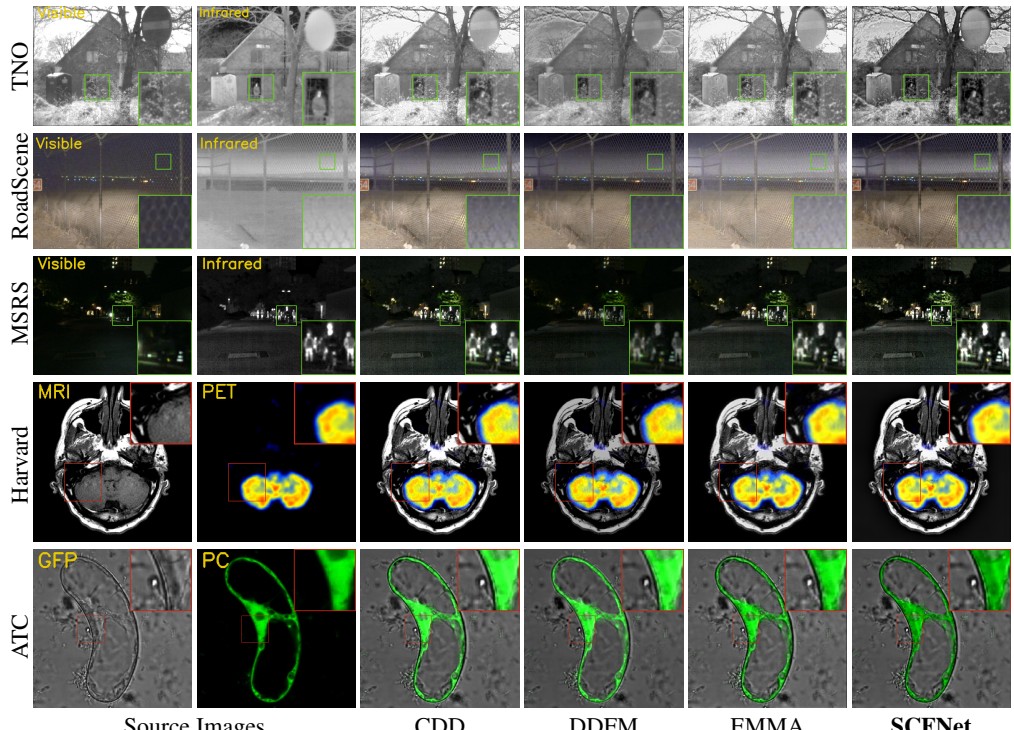

Figure 5: Performance Comparison of MMIF Methods.

Table 2: Quantitative results on TNO dataset of IVF and Harvard dataset of MIF.

| Methods | EN↑ | SD↑ | SF↑ | AG↑ | Qbaf↑ | VIF↑ | SSIM↑ | EN↑ | SD↑ | SF↑ | AG↑ | Qbaf↑ | VIF↑ | SSIM↑ |
|---------|-----|-----|-----|-----|-------|------|-------|-----|-----|-----|-----|-------|------|-------|
| | IVF on TNO dataset | | | | | | | MIF on Harvard Medical dataset | | | | | | |
| TarD | 7.02 | 49.89 | 8.68 | 3.81 | 0.28 | 0.54 | 0.83 | 3.91 | 55.94 | 21.62 | 4.04 | 0.42 | 0.57 | 0.82 |
| DeF | 7.06 | 40.70 | 8.21 | 3.76 | 0.43 | 0.64 | 0.92 | 4.26 | 52.49 | 24.08 | 4.32 | 0.60 | 0.62 | 1.21 |
| MURF | 6.93 | 34.95 | 8.37 | 3.45 | 0.37 | 0.55 | 0.87 | 4.42 | 36.35 | 24.18 | 5.98 | 0.56 | 0.37 | 0.94 |
| MetaF | 6.84 | 33.37 | 12.05 | 4.80 | 0.49 | 0.44 | 1.00 | - | - | - | - | - | - | - |
| CDD | 7.12 | 46.00 | 13.15 | 4.90 | 0.54 | **0.77** | 1.03 | 4.06 | **77.26** | 24.97 | 6.37 | 0.66 | 0.63 | 1.43 |
| DDFM | 7.06 | 49.71 | 10.45 | 4.19 | 0.47 | 0.71 | 0.98 | 4.21 | 62.81 | 22.43 | 6.11 | 0.59 | 0.63 | 1.16 |
| EMMA | 7.16 | 46.78 | 11.67 | 4.74 | 0.42 | 0.61 | **1.27** | 4.66 | 69.30 | 23.10 | 6.44 | 0.55 | 0.61 | 1.50 |
| **SCFNet** | **7.37** | **52.42** | **14.95** | **6.01** | **0.57** | 0.77 | 1.18 | **4.71** | 71.30 | **25.78** | **6.62** | **0.72** | **0.72** | **1.51** |

performance underscores the robust generalization capabilities of our method across diverse fusion scenarios. The results on MSRS (Tang et al., 2022b) dataset validate the stability and applicability of SCFNet for practical scenarios with diverse degradations and source variations. Additionally, improvements in SD metrics indicate that SCFNet significantly enhances contrast, particularly in challenging low-light conditions. By effectively consolidating multi-modality details across changing conditions, SCFNet demonstrates reliable fusion capabilities amid real-world challenges.

As illustrated in the top two rows of Fig. 5, SCFNet demonstrates superior integration of complete content information from both infrared and visible images, particularly in the highlighted areas, where people and wire mesh structures in the dark are effectively represented, and scene details are preserved. In low-light conditions, where existing methods struggle to maintain the intricate texture structure, SCFNet generates fused images with clearer details and more distinguishable foreground objects and backgrounds with abundant contour information. This further demonstrates that SCFNet demonstrates particular strengths in enhanced vision across challenging imaging degradation.

## 4.3 MEDICAL IMAGE FUSION AND BIOLOGICAL IMAGE FUSION

**Medical Image Fusion.** Following Zhao et al. (2023b), the trained IVF model, without fine-tuning, is directly evaluated on the Harvard Medical (website) dataset. This dataset comprises 21 pairs

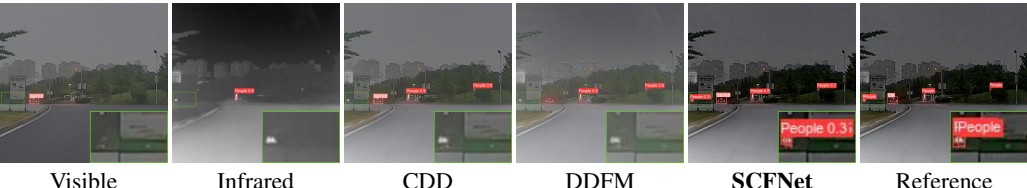

| Visible | Infrared | CDD | DDFM | **SCFNet** | Reference |

Figure 6: Visualizations of object detection on M³FD dataset.

Table 3: Mean average precision (mAP% ↑) values for semantic segmentation on the MSRS dataset and mean Intersection-over-Union (mIOU% ↑) values for object detection on the M³FD dataset.

| | Detection on M³FD dataset | | | | | | | Segmentation on MSRS dataset | | | | | | | | | |
|---|---|---|---|---|---|---|---|---|---|---|---|---|---|---|---|---|---|
| | *Bus* | *Car* | *Lam.* | *Mot.* | *Peo.* | *Tru.* | mAP@0.5 | *Unl.* | *Car* | *Per.* | *Bik.* | *Cur.* | *CS.* | *GD.* | *CC.* | *Bu.* | mIOU |
| Infrared | 78.8 | 88.7 | 70.2 | 63.4 | 80.9 | 65.8 | 74.6 | 90.5 | 75.6 | 45.4 | 59.4 | 37.2 | 51.0 | 46.4 | 43.5 | 50.2 | 55.4 |
| Visible | 78.3 | 90.7 | 86.4 | 69.3 | 70.5 | 70.9 | 77.7 | 84.7 | 67.8 | 56.4 | 51.8 | 34.6 | 39.3 | 42.2 | 40.2 | 48.4 | 51.7 |
| TarD | 81.3 | **94.8** | 87.1 | 69.3 | 81.5 | 68.7 | 80.5 | 97.1 | 79.1 | 55.4 | 59.0 | 33.6 | 49.4 | 54.9 | 42.6 | 53.5 | 58.3 |
| DeF | 82.9 | 92.5 | 87.8 | 69.5 | 80.8 | 71.4 | 80.8 | 97.5 | 82.6 | 61.1 | 62.6 | 40.4 | 51.5 | 48.1 | 47.9 | 54.8 | 60.7 |
| MURF | 81.3 | 92.6 | 86.5 | 70.8 | 80.2 | 69.9 | 80.2 | 97.2 | 81.4 | 62.0 | 60.9 | 39.7 | 52.3 | 55.5 | 46.8 | 56.1 | 61.3 |
| MetaF | 83.0 | 93.4 | 87.3 | 74.8 | 81.6 | 68.8 | 81.5 | 97.3 | 81.6 | 61.2 | 62.1 | 37.2 | 52.9 | 59.8 | 46.2 | 56.2 | 61.6 |
| CDD | 81.8 | 92.9 | 87.6 | 72.8 | 81.8 | 72.9 | 81.6 | 97.8 | 82.5 | 63.2 | 62.2 | 40.8 | 52.7 | 56.2 | 45.3 | **58.7** | 62.2 |
| DDFM | 82.2 | 93.2 | 87.6 | 68.4 | 81.0 | 71.3 | 80.6 | 97.4 | 82.5 | 60.4 | 62.0 | 41.7 | 52.9 | 56.2 | 46.3 | 53.7 | 61.2 |
| SegM | 81.8 | 93.1 | 86.8 | 72.3 | 79.9 | 70.9 | 80.8 | 97.6 | 84.6 | 64.8 | 63.6 | 40.2 | 52.9 | 59.9 | 49.4 | 56.2 | 63.2 |
| EMMA | 83.2 | 93.5 | 87.7 | 77.7 | 82.0 | 73.5 | 82.9 | 97.6 | 84.0 | 65.2 | 63.1 | **42.4** | 53.6 | 60.2 | 50.5 | 56.3 | 63.7 |
| **SCFNet** | **85.1** | 94.2 | **89.9** | **80.1** | **83.6** | **74.5** | **84.6** | **97.9** | **85.9** | **70.3** | **66.5** | 41.3 | **55.0** | 60.1 | 55.7 | 58.5 | **65.7** |

of MRI-CT, 42 pairs of MRI-PET, and 73 pairs of MRI-SPECT images. The quantitative results are presented in Tab. 2. Compared to CDD (Zhao et al., 2023b), which achieves a higher SD but sacrifices performance on other metrics, SCFNet outperforms other MMIF methods across most metrics, further validating its strong generalization capability. The fusion results illustrated in the third line of Fig. 5 demonstrate that our method preserves the detailed structural features of MRI as a foundation while incorporating color information from PET, whereas other methods obscure some tissue information. Combining the strengths of different modalities, this comprehensive fusion result provides enhanced visualization that better facilitates medical diagnosis.

**Biological Image Fusion.** We extensively evaluate our method for the BIF task on the ATC (Koroleva et al., 2005) dataset, which contains 128 image pairs of green fluorescent protein (GFP) and phase contrast (PC) images. The fusion network is retrained using 85 pairs for training, 18 pairs for validation, and 25 pairs for testing. Quantitative results are shown in *Supplementary*. These results clearly demonstrate SCFNet achieves superior performance across all metrics, validating our efficacy. The bottom line of Fig. 5 demonstrates that SCFNet preserves the structural integrity of the cytoplasm, especially in areas where other methods obscure the cell wall, while effectively suppressing noise. This balance is crucial for biological imaging applications where clarity and detail retention are paramount for accurate analysis and interpretation.

## 4.4 DOWNSTREAM APPLICATIONS

**Object Detection.** The evaluation of our SCFNet for object detection is conducted using the M³FD (Liu et al., 2022) dataset. This dataset is composed of 4,200 infrared and visible image pairs. The detection network YOLOv5 (Jocher, 2020) is retrained using 3,360 pairs for training, 420 pairs for validation, and 420 pairs for testing. This network is employed to detect six categories: *buses*, *cars*, *lamps*, *motorcycles*, *people* and *trucks*. As shown in Tab 3, our method achieves the top five results across all evaluation metrics. Fig 6 shows our SCFNet effectively helps detect distant people who are easily overlooked due to fog. This demonstrates that SCFNet can produce images more suitable for detection by enhancing contrast and highlighting foregrounds, maintaining strong performance even with degraded source images.

**Semantic Segmentation.** On the MSRS (Tang et al., 2022b) dataset, we retrain the segmentation network, DeeplabV3+ (Chen et al., 2018), for multi-modality scene segmentation task as (Zhao

Table 4: Ablation results of VIF on TNO and RoadScene datasets.

| Configurations | IVF on TNO dataset | | | | | IVF on RoadScene dataset | | | | |
|---|---|---|---|---|---|---|---|---|---|---|
| | EN↑ | SD↑ | SF↑ | Qbaf↑ | VIF↑ | EN↑ | SD↑ | SF↑ | Qbaf↑ | VIF↑ |
| FPE → Restor. | 7.29 | 51.90 | 14.16 | 0.56 | 0.75 | 7.42 | 53.31 | 17.14 | 0.54 | 0.70 |
| w/o SNF | 7.06 | 46.53 | 12.38 | 0.49 | 0.71 | 7.15 | 50.02 | 14.43 | 0.50 | 0.63 |
| w/o SAF | 7.15 | 49.57 | 13.70 | 0.52 | 0.73 | 7.30 | 52.13 | 15.89 | 0.51 | 0.65 |
| w/o $\mathcal{L}_{AR}$ | 7.20 | 47.81 | 12.65 | 0.54 | 0.75 | 7.22 | 51.61 | 14.50 | 0.53 | 0.68 |
| w/ $\mathcal{L}_{AR}^{-}$ | 7.26 | 51.62 | 14.08 | 0.51 | 0.73 | 7.28 | 52.00 | 17.34 | 0.52 | 0.65 |
| **Full (SCFNet )** | **7.37** | **52.42** | **14.95** | **0.57** | **0.77** | **7.55** | **55.29** | **18.32** | **0.56** | **0.72** |

et al., 2024; Liu et al., 2023b). This network is employed to manage nine categories of pixel-level labels, including *Unlabeled (background)*, *Car*, *Person*, *Bike*, *Curb*, *Car stop*, *Guardrail*, *Color cone*, *Bump*. From Tab. 3, performances as measured by the Intersection over Union (IoU) indicate that our method enhances the overall clarity of segmentation because fusion results produced by SCFNet possess more complete edges and details. More qualitative results are shown in *Supplementary*.

## 4.5 ABLATION STUDY

We construct experiments from the proposed SCFNet to perform analyses quantifying the impact of components. The ablation results are summarized in Tab. 4. To analyze the effectiveness of each component in our network, we design the following baseline model. (a) "FPE → Restor." : replace FPE blocks in the encoders with the recent CNN/Transformer-based Restormer blocks (Zhao et al., 2023b). Incorporating FPE blocks for feature frequency extraction results in significant performance gains. (b) "w/o SNF": replace SNF modules with concatenation and convolutions containing similar parameters. The result demonstrates the SNF effectively facilitates the fusion of style discrepancies between different modalities. (c) "w/o SAF": replace SAF with layers concatenation and convolutions with similar parameters. Better results with SAF are credited to the feature domain alignment and regularization that avoids fusion ambiguity.

Additionally, SAF supports flexible alignment of fused features to different source modalities without the need for retraining, making SAF more robust across varying modalities. As demonstrated in Fig. 7, aligning to different modalities produces high-quality fusion results with distinct visual characteristics, effectively meeting diverse application needs.

| *V* | *I* | Aligned *V* | Aligned *I* |

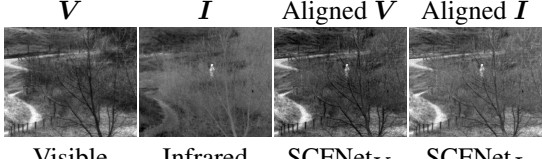

| Visible | Infrared | SCFNet$_V$ | SCFNet$_I$ |

Figure 7: The fusion results of SAF align different source modalities effectively.

Next, we design the following training strategies to analyze the proposed loss function. (d) "w/o $\mathcal{L}_{AR}$": remove the rescaled adjustment $\mathcal{R}$ in $\mathcal{L}_{AR}$, and directly use the original input images as supervision. The result indicates that $\mathcal{L}_{AR}$ is crucial for optimal performance. Even without $\mathcal{L}_{AR}$, SCFNet still demonstrates robust fusion capabilities, which can be attributed to our effective network design. (e) "w/ $\mathcal{L}_{AR}^{-}$": eliminate the domain identity learning loss term by setting the $\gamma$ in Eq. 11 to 0. The performance degradation proves that guiding the network to learn specific image domain information is beneficial for constraining the network to avoid distortion and obtain high-quality fused images. More analyses provided in *Supplementary* demonstrate that our proposed $\mathcal{L}_{AR}$ offers more comprehensive scene information supervision while preserving image priors.

## 5 CONCLUSION

In this work, we propose SCFNet, a novel style-coherent multi-modality fusion model, to address the challenge of integrating heterogeneous modalities in MMIF. To complement missing scene information from uncollected modalities, SNF enhances features by merging style-normalized representations. SAF aligns cross-modal fused features to a designated modality, ensuring stylistic consistency. Additionally, to improve the integrity and completeness of self-supervision, we employ an adaptive reconstruction loss function that linearly transforms source inputs to enforce mapping to a specific domain with image priors. Experiments demonstrate that our approach outperforms existing MMIF methods and shows strong potential for downstream applications.

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
