# STYLE-COHERENT MULTI-MODALITY IMAGE FUSION (SUPPLEMENTAL MATERIAL)

## 1 FOURIER PRIOR EMBEDDED BLOCK

The Fourier Prior Embedded (FPE) block is designed to capture frequency information of multimodal features through two main processes: Fourier Spatial Interaction (FSI) and Fourier Channel Interaction (FCI), as illustrated in Fig. 1. Specifically, for a given feature $X$, the fast Fourier transform is first applied, resulting in real and imaginary components denoted as $\text{Real}(X)$ and $\text{Im}(X)$, respectively. The Fourier Spatial Interaction (FSI) process then operates on these components independently to maintain fidelity in frequency manipulation. This process can be formulated as:

$$\text{Real}(X'_S) = \text{ReLU}(\text{DConv}(\text{Real}(X))), \text{Im}(X'_S) = \text{ReLU}(\text{DConv}(\text{Im}(X))), \quad (1)$$

where $\text{ReLU}(\cdot)$ is the ReLU function, and $\text{DConv}(\cdot)$ indicates the depth-wise convolution. Following FSI, the spatially enhanced feature is merged with the original spatial feature through concatenation and convolution, denoted as $X_S$.

Following the Fourier Spatial Interaction (FSI) process, the spatially enhanced feature $X_S$ undergoes further refinement through FCI. FCI enhances the channel-wise details of the feature frequencies using point-wise convolution, which is formulated as follows:

$$\text{Real}(X'_C) = \text{ReLU}(\text{Conv}_1(\text{Real}(X_S))), \text{Im}(X'_C) = \text{ReLU}(\text{Conv}_1(\text{Im}(X_S))), \quad (2)$$

where $\text{Conv}_1$ indicates the $1 \times 1$ convolution. Finally, a similar merging process occurs after the FPI, yielding the output of the FPE module, which achieves global modeling for both spatial and channel dimensions. Overall, FPE effectively extracts frequency information from the modality features of each branch. Subsequently, these features are enhanced across modalities using SNF. The enhanced features then serve as input for the next FPE block.

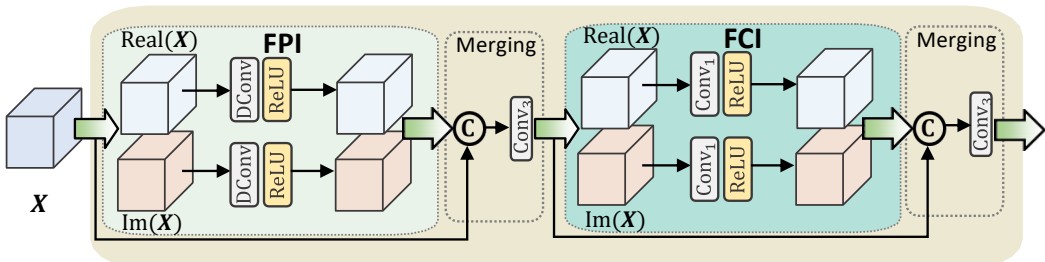

Figure 1: The architecture of FPE.

## 2 MORE TRAINING DETAILS

For the IVF task, we directly quote results on the MSRS [1] and RoadScene[2] datasets from (Zhao et al., 2023a; 2024). By training the available codes, we obtain results on the TNO[3] dataset.

---

[1] https://github.com/Linfeng-Tang/MSRS
[2] https://github.com/hanna-xu/RoadScene
[3] https://figshare.com/articles/dataset/TNOImageFusionDataset/1008029

For the MIF task, we obtain results on the Harvard medical [4] dataset by applying the trained IVF models to the MIF task without fine-tuning.

For the BIF task, we use the released code to retrain models with the same data settings as ours, obtaining results on the ATC [5] dataset. The training data is cropped to patches of size $256 \times 256$.

For downstream applications, we follow (Zhao et al., 2023a; 2024) for implementing semantic segmentation and object detection. For semantic segmentation, we directly utilize trained IVF models to obtain fusion images on the MSRS dataset. Following the dataset splits in (Tang et al., 2022b), we retrain the DeeplabV3+ segmentation model with cross-entropy loss using suggested hyperparameter settings from the released code[6]. For object detection, We utilize trained IVF models and obtain fusion images on the $M^3FD$[7] dataset, where 3,360 images are used for training, 420 images for validation, and 420 images for testing. We retrain the YoLo+ detection model using suggested hyperparameter settings from the released code[8].

# 3 BIOLOGICAL IMAGE FUSION

From Tab. 1, it can be observed that our proposed method outperformed the other MMIF methods in most of the evaluation metrics. This comprehensively demonstrates the superior performance of our proposed style-coherent fusion model (SCFNet). Fig. 2 shows that our SCFNet effectively captures cellular structural features from PC while suppressing noise from GFP.

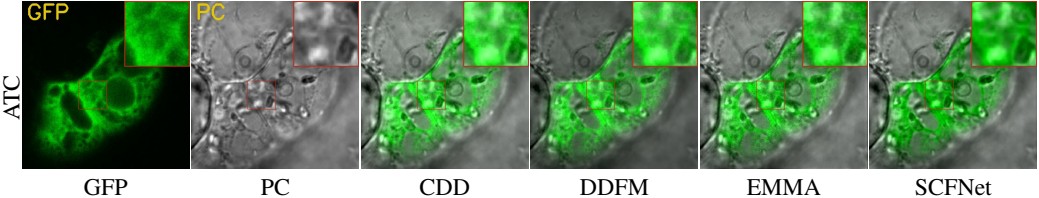

GFP  PC  CDD  DDFM  EMMA  SCFNet

Figure 2: More visualizations of BIF on ATC (Koroleva et al., 2005) dataset.

Table 1: Quantitative results of BIF on ATC dataset.

| Methods | EN↑ | SD↑ | SF↑ | AG↑ | Qbaf↑ | VIF↑ | SSIM↑ |
|---------|-----|-----|-----|-----|-------|------|-------|
| TarD | 6.08 | 40.22 | 9.48 | 2.31 | 0.52 | 0.86 | 0.85 |
| DeF | 6.46 | 42.63 | 9.60 | 2.80 | 0.61 | 0.92 | 0.92 |
| MURF | 6.15 | 41.82 | 9.91 | 2.67 | 0.60 | 0.96 | 0.81 |
| CDDFuse | 6.70 | 48.38 | 12.56 | 3.73 | 0.63 | **1.05** | 1.00 |
| DDFM | 6.53 | 47.04 | 11.44 | 2.51 | 0.59 | 0.95 | 0.96 |
| EMMA | 6.71 | 49.13 | 12.58 | 3.76 | 0.58 | 0.97 | 1.04 |
| **SCFNet** | **6.82** | **51.34** | **14.01** | **4.04** | **0.64** | 1.01 | **1.15** |

# 4 MORE ANALYSES

## 4.1 ANALYSES OF STYLE-ALIGNMENT FUSION

We further analyze that SAF enables the selection of alignment across different modalities, including visible or infrared domains. It is important to note that SAF uses a well-defined source distribution to guide the diverse features into a unified domain, rather than directly defining the target domain.

---

[4] http://www.med.harvard.edu/AANLIB/home.html
[5] http://data.jic.bbsrc.ac.uk/gfp
[6] https://github.com/VainF/DeepLabV3Plus-Pytorch
[7] https://github.com/JinyuanLiu-CV/TarDAL
[8] https://github.com/ultralytics/yolov5

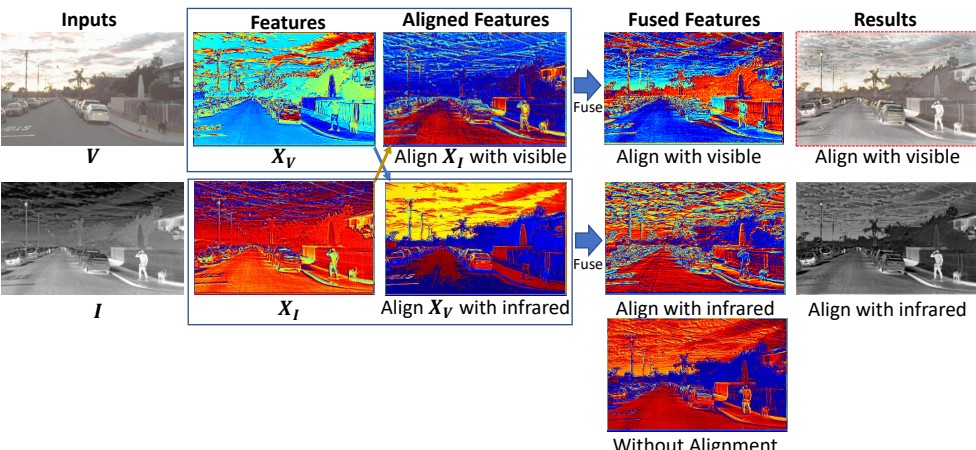

Figure 3: The visualization results of the feature alignment strategies based on style-alignment fusion module (SAF). It includes the alignment of features with the visible and infrared domains, as well as the fused results. The fused features with alignment retain more scene details. The result aligned with the visible domain exhibit superior visual performance.

Table 2: Ablation studies of the selection of alignment across different domains in SAF on Road-Scene dataset.

| SAF | EN ↑ | SF ↑ | Qbaf↑ | VIF ↑ | SSIM↑ |
|---|---|---|---|---|---|
| Align w/ infrared domain | **7.63** | **19.06** | 0.53 | 0.68 | 1.03 |
| Align w/ visible domain (Original) | 7.55 | 18.32 | **0.56** | **0.72** | **1.21** |

Compared to the channel-wise fusion of features without alignment, the features fused using SAF exhibit clearer details in the sky and vehicles. This improvement highlights the effectiveness of the SAF approach in preserving and enhancing critical scene elements. When SAF aligns the infrared domain, as shown in Fig. 3, the explicitly aligned infrared domain $X_V$ exhibits higher contrast on vehicles and more prominent landmarks. The resulting fusion preserves fine details, but some edges appear overly accentuated. When SAF aligns with the visible domain, the explicitly aligned $X_I$ retains complete information on thermal tasks and power lines. As illustrated in the third row of Fig. 3, the aligned fusion features preserve the complete scene details of the source modalities, differing only in the visual effects of the reconstructed images. The fusion result with visible domain alignment is more visually satisfactory, striking a better balance between preserving details and maintaining natural appearance.

From the results in Tab. 2, it is evident that although aligning with the infrared domain increase the information entropy of the fused image, the visual quality assessments including metrics such as Qbaf, VIF, and SSIM tend to significantly decrease. Therefore, we prioritize alignment with the domain that contains more information to avoid excessive adjustments that could deteriorate the visual quality of the fused image.

## 4.2 ANALYSES OF ADAPTIVE RECONSTRUCTION LOSS

Based on the ablation study in Sec. 4.5 of the main paper that compares different training strategies, we provide further visual comparisons to validate the effectiveness of our proposed adaptive reconstruction loss function.

First, we compare the image-level supervision signals $(\text{Max}(V, I))$ generated by traditional loss functions (Zhao et al., 2023a; Tang et al., 2022a) and our proposed loss function with supervision signals $(\text{Max}(\mathcal{R}(V), \mathcal{R}(I)))$. It is evident that the learnable rescaled function $\mathcal{R}$, defined in Eq.9 in the main paper, significantly enhances the utilization of multi-modality images to supervise the model, resulting in more comprehensive information, particularly for people obscured by smoke

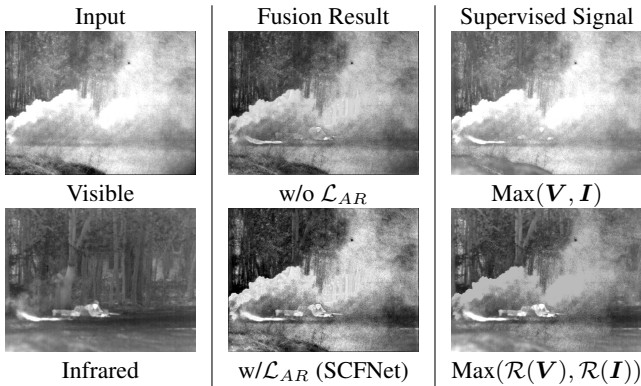

Figure 4: Visualizations of ablation studies for $\mathcal{L}_{AR}$. The supervision signals in $\mathcal{L}_{AR}$, compared to existing loss functions, guide our SCFNet in generating complete scene details with higher contrast.

Table 3: Ablation studies of $\beta$ in Eq.9 and operations in Eq.11 on RoadScene dataset.

| | Methods | EN↑ | SD↑ | SF↑ | Qbaf↑ | VIF↑ |
|---|---|---|---|---|---|---|
| $\beta$ in Eq.9 | Mean($I$) | 7.31 | 53.72 | 17.61 | 0.52 | 0.70 |
| | Learnable | 7.41 | 53.90 | 18.04 | 0.54 | 0.71 |
| | **Mean($V$) (Original)** | **7.55** | **55.29** | **18.32** | **0.56** | **0.72** |
| Operations in Eq.11 | Mean($\cdot$) | 6.67 | 48.5 | 15.13 | 0.53 | 0.66 |
| | Separation | 7.42 | 53.14 | 17.99 | 0.54 | 0.71 |
| | **Max($\cdot$) (Original)** | **7.55** | **55.29** | **18.32** | **0.56** | **0.72** |

and details of tress, while enhancing contrast and suppressing noise to ensure that the blurry, noisy visible image content is effectively presented. Additionally, the visualization of "w/o $\mathcal{L}_{AR}$" further demonstrates that fusion models trained without the $\mathcal{L}_{AR}$ loss are prone to degradation, resulting in low image contrast, making the person in smoke less noticeable. Therefore, the proposed adaptive reconstruction loss function addresses the absence of GT and effectively guides the model to generate high-quality images with complete information, facilitating further improvements in downstream applications such as detection.

We specifically analyze the $\beta$ of the main paper Eq.9, which is set to Mean($X$) to align $\hat{X}_F$ with the distribution of the visible image domain. The results in Tab. 3 indicate that setting $\beta = $ Mean($I$), denoted as "Mean($I$)", significantly reduces the effectiveness of fusion. This reduction in performance is primarily attributable to alignment conflicts with our proposed SAF which is oriented toward the visible domain. We also set $\beta$ learnable and constrain it within [Mean($I$), Mean($V$)], denoted as "Learnable". Due to the negative impacts of the alignment conflict being mitigated, this results in a less pronounced reduction in performance. However, introducing too many learnable variables for the supervision signal leads to training instability. Consequently, fixing $\beta$ at Mean($V$) enables a more effective and stable adjustment of source images to align with the visible domain.

Additionally, in the main paper Eq.11, We follow Zhao et al. (2023a); Tang et al. (2022a) by utilizing the maximum pixel values as the supervision signal to enhance the overall clarity of the fused image. As shown in the following Tab. 3, replacing with the mean operation, denoted as "Mean($\cdot$)", leads to a significant decrease in performance and loss of a substantial amount of detail. Separate supervisions with $R(I)$ and $R(V)$ as Xu et al. (2022); Zhao et al. (2020); Xu et al. (2020a), denoted as "Separation". It also leads to decreased performance, as the fusion model tends to blend the two images with a smoothing effect.

## 5 LIMITATIONS

Similar to existing methods (Liu et al., 2023; Sun et al., 2022; Tang et al., 2022a; Zhao et al., 2024; 2023b; Yi et al., 2024), the proposed SCFNet primarily focuses on geometrically calibrated multi-

modality images. However, in some cases, capturing simultaneously from the same scene can be challenging due to differences in sensor perspectives and positions, often resulting in misalignments in the collected multi-modality images. Effectively training our SCFNet with misaligned multi-modality images will be the focus of our future work.

# 6 MORE VISUALIZATIONS

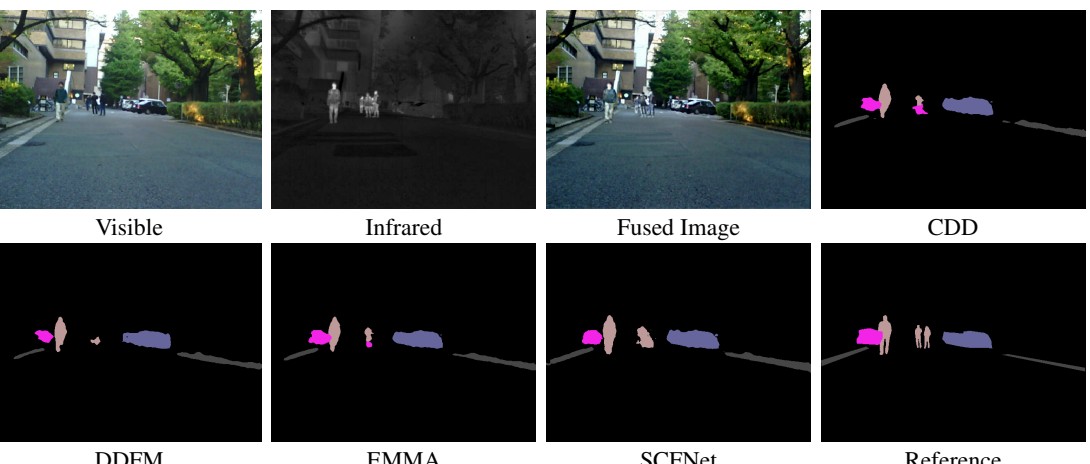

Figure 5: Visualizations of semantic segmentation on MSRS (Tang et al., 2022b) dataset.

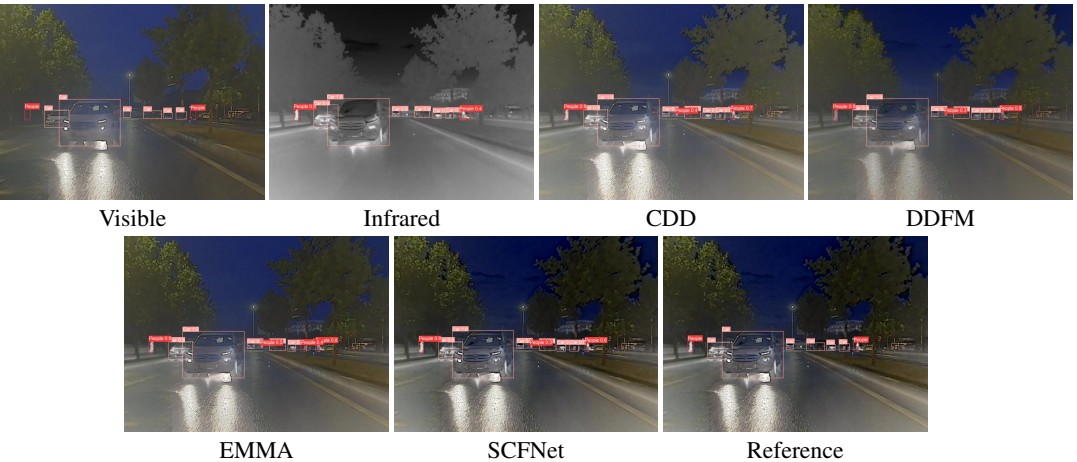

Figure 6: More visualizations of object detection on M³FD (Liu et al., 2022) dataset.

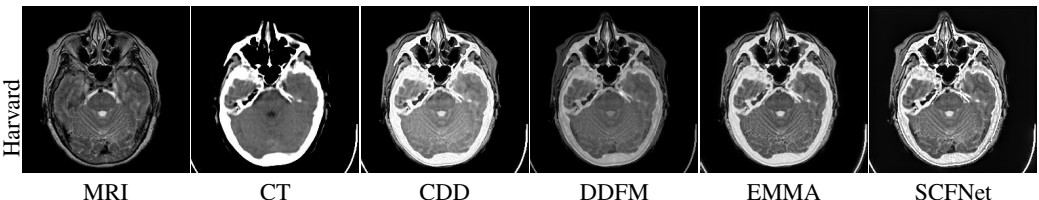

Figure 7: More visualizations of MIF on Harvard Medical (website) dataset.

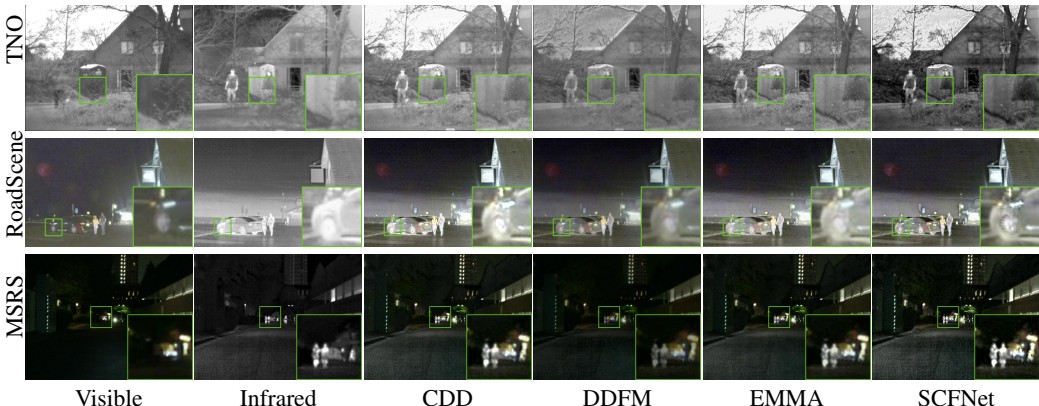

Figure 8: More visualizations of IVF on TNO (Toet & Hogervorst, 2012), RoadScene (Xu et al., 2020b), and MSRS (Tang et al., 2022b) datasets.