# OpenReview forum: "Style-Coherent Multi-Modality Image Fusion"
_ICLR.cc/2025/Conference — Submitted to ICLR 2025_

### Official Review · Reviewer_ooZi · 2024-10-25

**Soundness:** 3
**Presentation:** 3
**Contribution:** 3
**Rating:** 6
**Confidence:** 4

**Summary:**

In this paper, the authors study the problem of fusing images from multiple modalities for various downstream tasks. To this end, the authors propose a deep network to handle the discrepancies between different modalities. In this network, the authors first split the amplitude and the phase in the frequency domain, leveraging the observation that style (modality-specific details) are preserved in the amplitude where other details (content) are represented in the phase component. The network first style-normalizes features from both modalities and then uses a learnable alignment to obtain a unified representation in the visible domain.

The results on several benchmarks suggest significant improvements over the state of the art.

**Strengths:**

+ Multi-modal image fusion is a fundamental step in many applications.
+ The proposed approach of separating style and content is sound and promising.
+ The results are convincing.

**Weaknesses:**

1. I sense that the authors specifically avoided the use of the term disentanglement. In the disentanglement literature, people did introduce different methods for disentangling content and style for various applications. I believe positioning the paper with that literature would have been valuable. A quick Google search reveals some studies using disentanglement for some multimodal tasks, though with non-visual modalities.

2. Figure 1: I am not convinced with the visual results provided. The only difference I see is that the proposed method produces slightly sharper reconstructions. This does not necessarily entail that the style discrepancy is a major issue. The example in Figure 7 is more convincing.

3. Many crucial details are left unclear.

3.1. Figure 2: It is not clear how Merging is different from Summation or Concatenation. The figure/caption should state what FPE stands for.

3.2. "the twin encoder branches share the same structure and parameters." => This should be justified a bit.

3.3. "the degree of style modification is gradually adjusted by introducing learnable parameters" => How do we ensure that this is gradual if the parameters are learnable.

3.4. Eq 7: Not clear why pooling is required here or why there is a need for a spatial squeeze operation. Moreover, it is not justified why maxpool has to be combined with avgpool.

3.5. Eq 8: What's being performed here is not explained properly.

3.6. Entropy-aware alpha: Not clear why providing bounds on a variable enforces information entropy.

3.7. Eq 11: This overall loss formulation should have been explained in more detail. There are several unclear bits. Why do we use Max(R(V), R(I)) for similarity loss? Why is there no (f(V, I)-I) or (f(I, I)-I) term in the equation?

4. The paper should provide analysis on alpha and the parameters in Eqs 5 and 6.

Minor comments:
- "Fourier Prior Embedded" => "Fourier Prior Embedding".
- "we perform a content replacement" => "we replace content".
- Eq 9: I suppose it is better to use X instead of I.

**Questions:**

Please see Weaknesses.

---

> ### Author Response · Authors · 2024-11-22
> **To Reviewer ooZi**
>
> ``1. Disentanglement for some non-visual multi-modality tasks. ``
>
> We add relevant Disentanglement methods for multi-modality tasks in the main paper Sec.2. It highlights the contributions of SCFNet in fusing multi-modal images based with distinct visual characteristic on style-content decomposition.
>
> ``2. About Fig.1.``
>
> We add fused features in the main paper Fig.1. It reveals that fused features without style adjustments are smooth and lack details. Our SCFNet, by fusing style-coherent features, better captures details such as bridge structures and tree edges. Fig. 7 shows SCFNet producing fused images in various styles while preserving complete scene details, highlighting the effectiveness of our SCFNet in integrating content with style modulation.
>
> ``3. About Fig.2.``
>
> The merging operator consists of concatenation followed by a convolutional layer, as shown in the supplementary Fig.1. We also add the description of FPE in the caption of Fig.2.
>
> ``4. About the shared twin encoder branches.``
>
> The reason the twin encoder branches share the same structure and parameters is to reduce complexity and potential discrepancies that might arise from having separate configurations for each modality. We add this into the main paper.
>
> ``5. The degree of style modification is gradually adjusted. ``
>
> We conduct an analysis of the learnable $\lambda^1$ and $\lambda^2$ in Eq.5 and Eq.6.  As shown in the following Tab.1, at the initial level of encoders, SNF tends to preserve more of the source modality style, maintaining the uniqueness of each modality. As the level deepens, the model increasingly relies on the fused features to enrich the representation of content. Furthermore, since the fused feature is encouraged to align with the visible image domain, $\lambda^1$ for visible tends to preserve the source modal amplitude more than $\lambda^2$ for infrared.  We will include this analysis in the supplementary materials for the revision.
>
> Table1:Parameters $\lambda$ of SNF in IVF task.
>
> | Level | $\lambda^1_v$ | $\lambda^2_v$ | $\lambda^1_i $ | $\lambda^2_i$ |
> |-|-|-|-|-|
> | 1 | 0.62 | 0.38| 0.46 | 0.54  |
> | 2  | 0.45 | 0.55 | 0.37  | 0.63|
> | 3 | 0.33  | 0.67 |  0.09  | 0.91 |
>
> ``6. Pooling operations in Eq.7.``
>
> $W$ is learned to fuse style-normalized features.  Through spatial squeezing, the fusion process minimally alters the features, thus preserving essential details and reducing computational load.  The standard deviation of the fused features is adjusted using $W$ with $\text{std}(X_V)$ for alignment.  By employing max pooling, the most prominent features are accentuated, whereas average pooling offers a comprehensive overview of features.  This approach balances the focus on both critical and general details for distribution adjustment, optimizing the integration of infrared and visible features.
>
> ``7. About Eq.8. ``
>
> Eq. 8 is used to adjust the distributional characteristics of the style-normalized fused feature to align with the visible feature $\hat{X}_V^L$. The learnable weight $W$ is utilized to channel-wise fuse $SN(X_V)$ and $SN(X_I)$, and then align the fused feature with the variance of the style-normalized $\hat{X}_V^L$. The mean of the style-normalized fused feature is adjusted to match the mean of $\hat{X}_I^L$. This process ensures that the fused output retains crucial information from modalities, optimized for the characteristics of the visible domain. We revise the main paper for more clarity.
>
> ``8. About entropy-aware \alpha. ``
>
> An excessively large or small entropy-aware $\alpha$ damages the image priors of the supervision signal.   Such constraints make the modal image with smaller variances effectively align with other source images, while ensuring that the adjusted supervision signal tends toward consistency, thereby ensuring the integrity of the supervision signal. We conduct an additional experiment by removing the constraints on $\alpha$. The results show a significant performance decrease on RoadScene dataset, as shown in the following Table 3. This is due to excessive modulation of the source images, which cannot provide effective supervision, thereby demonstrating the necessity of the constraint.
>
> Table 3: Parameters $\alpha$  of adaptive reconstruction loss on RoadScene dataset.
>
> | $\alpha$         | EN    | SF   | Qbaf | VIF  | SSIM |
> |------------------|-------|------|------|------|------|
> | w/o constrain| 5.97  | 14.30 | 0.41 | 0.57 | 0.78 |
>  | **w/ constrain (Original)** | **7.55** | **18.32** | **0.56** | **0.72** | **1.21** |

---

> ### Author Response · Authors · 2024-11-22
> **To Reviewer ooZi**
>
> ``9.Use Max(R(V), R(I)) for similarity loss.``
>
> We follow [1,2],  using the maximum pixel values as the supervision signal in Eq. 11. As detailed in the supplementary material, Section 4.2, we further analyzed the operations used to form the supervision signals in the loss function. As shown in the following Table 4, replacing the current max operation with a mean operation, or using separate supervisions for $R(I)$ and $R(V)$, both led to decreased performance. The key reason is that when using the mean or separate supervisions, the network tends to blend the two input signals too smoothly. This excessive blending results in a loss of important content details in the fused output. In contrast, the max operation better preserves the distinct content characteristics from the input modalities, preventing this detail loss.
>
> Table 4: Ablation studies of operations in Eq.11 on RoadScene dataset.
>
> | Methods               | EN ↑  | SD ↑   | SF ↑   | Qbaf ↑ | VIF ↑  |
> |-----------------------|-------|--------|--------|--------|--------|
> | $\text{Mean}(\cdot)$           | 6.67  | 48.50  | 15.13  | 0.53   | 0.66   |
> | $\text{Separation}$            | 7.42  | 53.14  | 17.99  | 0.54   | 0.71   |
> | **$\text{Max}(\cdot)$ (Original)** | **7.55** | **55.29** | **18.32** | **0.56** | **0.72** |
>
> [1] Correlation-driven dual-branch feature decomposition for multi-modality image fusion. CVPR, 2023.
>
> [2] Image fusion in the loop of high-level vision tasks: A semantic-aware real-time infrared and visible image fusion network. Inf. Fusion, 2022.
>
> ``10. About (f(V, I)-I) or (f(I, I)-I).``
>
> To avoid ambiguity during the fusion process, Our SCFNet prioritizes the domain with more information, the visible domain. The network learns the mapping of the visible image domain. Incorporating identity learning in the visible domain enhances the ability to capture visible features. However, introducing identity learning into the infrared image domain creates confusion during feature fusion, subsequently affecting the clarity of the fused images.
>
> ``11. Minor comments.``
>
> We revise Eq. 9 and correct typos in the main paper.

---

### Official Review · Reviewer_Exws · 2024-10-28

**Soundness:** 3
**Presentation:** 3
**Contribution:** 2
**Rating:** 5
**Confidence:** 4

**Summary:**

This paper presents the Style-coherent Content Fusion Model (SCFNet) for Multi-Modality Image Fusion (MMIF), addressing the challenges posed by significant style discrepancies between images from different sensors. The proposed model utilizes a dual-branch encoder architecture, incorporating a Fourier Prior Embedded (FPE) block, a Style-Normalized Fusion (SNF) module, and a Style-Alignment Fusion (SAF) module to enhance content representation and align features across modalities. An adaptive reconstruction loss function is introduced to improve content supervision and detail retention during fusion.

**Strengths:**

1. This paper aligns heterogeneous modal features to a shared and unified fusion space instead of directly fusing them, which is reasonable to reduce the differences between modalities.
2. The performance of this paper seems better compared to some related SOTA works.

**Weaknesses:**

1. The core of this paper is to align heterogeneous modal features to a shared latent fusion space to reduce inter-modal differences, which is reflected in Equation 8. However, there is a lack of theoretical analysis and further experimental validation regarding the rationale behind the design of the modal alignment method and its varying impacts. In particular, it needs to be analyzed through experiments whether aligning infrared features to the visible light domain will lead to the loss of certain infrared detail information, requiring an examination of the information retention during the alignment process.
2. Currently, there is a substantial amount of research focusing on both modal consistency and modal heterogeneity, such as CDDFuse [1], which employs fusion methods to address modal differences. Earlier, DRF [2] utilized a style transfer approach by separating scene and attribute representations for image fusion. The paper lacks a thorough comparative analysis of this work with existing similar studies and do not provide enough experimental comparisons with other similar methods.
3. As to the method, this paper mainly combines and improves existing approaches in the design of key components. For example, the design of the FPE and SNF modules draws on previous works, which, while beneficial for the research, offers limited contributions in terms of innovation.
4. This paper proposes adaptive reconstruction loss as one of the innovations, with the Equations 9-11, but the rationale and effectiveness lack explanation and validation. First, how should the hyperparameter $\beta$ in Equation 9 be set, as it is very important. Second, in Equation 11, why use $\max(R(V), R(I))$ — is $\max$ the optimal choice?
5. In the experiment section, the training set used by the authors follows the settings from [1] and [3]. In Table 2, the results of the CDD method on the TNO Dataset are consistent with [1], so the results of the same comparison methods, TarD and DeF, on the TNO Dataset should also match those in [1]. Why is there a discrepancy here?

[1] "CDDFuse: Correlation-driven dual-branch feature decomposition for multi-modality image fusion." Proceedings of the IEEE/CVF Conference on Computer Vision and Pattern Recognition, 2023.

[2] "DRF: Disentangled representation for visible and infrared image fusion." IEEE Transactions on Instrumentation and Measurement, 2021.

[3] "Equivariant multi-modality image fusion." Proceedings of the IEEE/CVF Conference on Computer Vision and Pattern Recognition, 2024.

**Questions:**

1. Please provide further ablation experiments on Equation 8, such as thoroughly analyzing the impact of different alignment strategies on the preservation of modal information and fusion effectiveness. Additionally, the paper should use visualization or quantitative methods to discuss and analyze the feature representation capability of the potential fusion space after alignment, as this is a core innovation of the study.
2. Related works need to be considered. I suggest that the authors include a more in-depth analysis of how their method compares to existing approaches in the introduction and related work sections. Additionally, since the paper extracts modal heterogeneity based on a decomposition approach, it is important to conduct experimental comparisons with previous related works. I suggest that the authors begin the discussion by referencing some earlier multimodal fusion papers based on decomposition approaches, such as DRF [2], DIDFuse [4], and LRRNet [5].
3. Please provide a sensitivity analysis experiment on the impact of different $\beta$ values on the fusion results. Additionally, an ablation experiment on the max operation in Equation 11 should also be conducted.
4. Please clarify the exact experimental settings used for comparison in the fifth Weakness and illustrate any differences from the settings used in CDDFuse [1].

[4] "DIDFuse: Deep image decomposition for infrared and visible image fusion," Proceedings of the International Joint Conference on Artificial Intelligence, 2020.

[5] "LRRNet: A novel representation learning guided fusion network for infrared and visible images." IEEE Transactions on Pattern Analysis and Machine Intelligence, 2023.

---

> ### Author Response · Authors · 2024-11-22
> **To Reviewer Exws**
>
> ``1. The core idea and Eq. 8.``
>
> Our core idea for fusion is reflected not only in Eq. 8 but also in SNF, where complete phase fusion of style-normalized multi-modal features is demonstrated in the main paper Eq. 4-6.
>
> **More Analysis**: Eq. 8 is used to adjust the style-normalized fused feature to align with the visible domain. The learnable $W$ is utilized to channel-wise fuse $SN(X_V)$ and $SN(X_I)$, and then align the fused feature with the  $\text{Std}(\hat{X}_V^L)$. The mean of the fused feature is adjusted to match  $\text{Mean}(\hat{X}_I^L)$. It ensures the fused feature is optimized for the characteristics of the visible domain. We revise the main paper for more clarity.
>
> **Experimental validation**:
>
> (1) For **feature alignment with a specific domain**, we present a visual comparison. Fig. 4 in the supplementary demonstrates that aligning modalities can reduce feature differences between modalities and does not lead to the loss of scene information.
>
> (2) For **the feature fusion process**, we conduct both visual and quantitative comparisons. As shown in Tab. 4 of the main paper, skipping the alignment and directly fusing results in significantly reduced performance, demonstrating the necessity of alignment for effective fusion. The comparison of visual results in supplementary Fig. 4 shows that fused features with alignment preserve complete scene information without over-smoothing.
>
> (3) For **SAF aligning different modalities**, we also conduct both visual and quantitative comparisons. The visualization results in supplementary Fig. 4 and the quantitative results in the following Table 1 highlight the advantages of selecting different modal domains for alignment. For IVF, although aligning with the visible domain provides more distinct details and contrast, maintaining excessively high contrast reduces visual performance. Therefore, prioritizing the information-abundant domain helps avoid the destruction of image priors caused by excessive modulation.
>
> Table 1: Ablation studies of the selection of alignment across different domains in SAF on RoadScene dataset.
>
> | SAF                           | EN ↑  | SF ↑   | Qbaf ↑ | VIF ↑  | SSIM ↑  |
> |-------------------------------|-------|--------|--------|--------|---------|
> | Align w/ infrared domain      | **7.63** | **19.06** | 0.53   | 0.68   | 1.03    |
> | Align w/ visible domain (Original) | 7.55  | 18.32   | **0.56** | **0.72** | **1.21** |
>
> ``2. About decomposition approaches.``
>
> In Sec. 2 of the main paper, we add related works and provide analysis. CDD[1] employs shared and specific multi-modal feature decomposition, followed by separate fusions of specific and shared features. DIDFuse[3] uses four distinct encoders to extract scenario features and attribute latent representations, and then fuse each set. However, these methods, including DRF[3], **do not address differences in feature characteristics, incomplete content complementarity, and ambiguity in the fusion outputs**.
>
> Specifically, (1) **Decomposition**: SCFNet is the first to leverage style learning to decompose into style-specific characteristics and content-invariant representations. (2) **Fusion**: With style-normalized features, SAF explicitly ensures that the content of the features maintains completeness and effective integration. (3) **Alignment**: SAF and the proposed loss function constrain the fused features and image domain alignment with a well-defined domain to ensure consistency.
>
> From the results in the main paper, our SCF outperforms the latest decomposition methods, CDD [1] and DeF [2], further demonstrating the effectiveness of our decomposition and fusion approach for MMIF.
>
> [1] Correlation-driven dual-branch feature decomposition for multi-modality image fusion. CVPR, 2023.
>
> [2] Fusion from decomposition: A self-supervised decomposition approach for image fusion. ECCV, 2022.
>
> [3] Disentangled representation for visible and infrared image fusion. IEEE Trans. Instrum. Meas.2021.
>
> ``3. The design of the FPE and SNF. ``
>
> To the best of our knowledge, our method is **the first to directly address multi-modal inconsistencies based on style-learning theory**. Our SNF module emphasizes integrating content and preserving details. To simultaneously handle different image features, our SNF module integrates content by **mapping heterogeneous features to a style-independent space**. It conducts **content fusion and replacement** to enhance dual-branch source content information and designs dynamic style adjustments to progressively refine feature expression and enhance details. These capabilities cannot be achieved through existing methods or previous works.
>
> Furthermore, **we do not claim FPE as our contribution**. FPE enhances the extraction of feature frequencies, which aids in subsequent feature alignment and fusion processes. It is not the contributor to resolving the modality-specific visual characteristic differences

---

> ### Author Response · Authors · 2024-11-22
> **To Reviewer Exws**
>
> ``4. About adaptive reconstruction loss with Eq. 9-11.``
>
> (1) Our proposed loss employs a learned linear rescaled function for the source input to form a unified and complete supervision signal. The visualization results in the supplementary Fig.3 demonstrate that the adaptive reconstruction loss function effectively guides the model to learn fusion, **forming a supervisory signal that encompasses complete scene information**.   Models trained under the supervision of our proposed loss function reconstruct efficient fused images, especially for obscured objects, such as individuals concealed by smoke.
>
> (2) **The analysis of the $\beta$** in the adaptive reconstruction function shows that setting it to $\text{Mean}(V)$ improves alignment with the visible domain. As shown in the following Table1, this is more effective than the learnable setting or $\text{Mean}(I)$, especially when combined with the style-adjustment fusion (SAF) module.
>
> Table 1: Ablation studies of $\beta$ in Eq.9 on RoadScene dataset.
> | Methods               | EN ↑  | SD ↑   | SF ↑   | Qbaf ↑ | VIF ↑  |
> |-----------------------|-------|--------|--------|--------|--------|
> | $\text{Mean}(I)$       | 7.31  | 53.72  | 17.61  | 0.52   | 0.70   |
> | Learnable             | 7.41  | 53.90  | 18.04  | 0.54   | 0.71   |
> | **$\text{Mean}(V)$ (Original)** | **7.55** | **55.29** | **18.32** | **0.56** | **0.72** |
>
> (3) We follow [1,3], using the maximum pixel values as the supervision signal in Eq. 11.  We further analyze **the operation used to form the supervision signals** in the loss function.  As shown in the following Table2, Replacing the current max operation with a mean operation, or using separate supervisions for $R(I)$ and $R(V)$, both led to decreased performance. The key reason is that when using the mean or separate supervisions, the network tends to blend the two input signals too smoothly. This excessive blending results in a loss of important content details in the fused output. In contrast, the max operation better preserves the distinct content characteristics from the source modalities, preventing smoothing.
>
> Table 2: Ablation studies of operations in Eq.11 on RoadScene dataset.
> | Methods               | EN ↑  | SD ↑   | SF ↑   | Qbaf ↑ | VIF ↑  |
> |-----------------------|-------|--------|--------|--------|--------|
> | $\text{Mean}(\cdot)$           | 6.67  | 48.50  | 15.13  | 0.53   | 0.66   |
> | $\text{Separation}$            | 7.42  | 53.14  | 17.99  | 0.54   | 0.71   |
> | **$\text{Max}(\cdot)$ (Original)** | **7.55** | **55.29** | **18.32** | **0.56** | **0.72** |
>
> [3] Image fusion in the loop of high-level vision tasks: A semantic-aware real-time infrared and visible image fusion network. Inf. Fusion, 2022.
>
> ``5. About results of TarD and DeF on TNO.``
>
> We follow the dataset configurations[1] to train SCFNet and other models with published codes, tuning hyperparameters to obtain promising results. We find TarD with loss function weight $\beta$=0.05 performs better, with results on other results consistent with [1]. Additionally, for DeF trained on the COCO dataset [4], we finetune it on our training dataset, achieving superior performance on the TNO dataset.
>
> [4] Microsoft coco: Common objects in context. ECCV, 2014.

---

> ### Author Response · Authors · 2024-11-29
> **Looking forward to your additional feedback**
>
> Dear Reviewer Exws,
>
> Thank you once more for your insightful review.
>
> As the author-reviewer discussion period is nearing its end, we would greatly value your feedback on whether our revisions and responses have adequately addressed your earlier concerns.
>
> Should you have any further questions or additional feedback, please let us know, and we will address them promptly.
>
> We appreciate your time and contributions.
>
> Warm regards,
>
> The Authors

---

### Official Review · Reviewer_vKhE · 2024-11-03

**Soundness:** 3
**Presentation:** 3
**Contribution:** 3
**Rating:** 6
**Confidence:** 4

**Summary:**

This paper presents a novel approach to Multi-Modality Image Fusion (MMIF) that addresses the issue of style discrepancies (e.g., saturation, resolution) in existing methods, which can obscure important features. The proposed model includes a style-normalized fusion module for more effective feature merging and a style-alignment fusion module to ensure consistency across modalities. An adaptive reconstruction loss enhances information preservation during the fusion process. Experimental results show that this method outperforms existing approaches, indicating strong potential for various image processing applications.

**Strengths:**

1.The style-coherent approach is applied to multimodal fusion field and valid to be effective.

2.The paper is well-written and easy to follow.

**Weaknesses:**

While the adaptive reconstruction loss is a key part of the proposed approach, the paper provides limited analysis on its impact compared to other loss functions. Further ablation studies focusing specifically on this component could strengthen the claims.

**Questions:**

See "Weaknesses"

---

> ### Author Response · Authors · 2024-11-22
> **To Reviewer vKhE**
>
> ``More analysis and ablation studies for proposed adaptive reconstruction losses.``
>
> More analysis with visualization results and quantitative results on the proposed adaptive reconstruction loss are provided in the supplementary Sec.4.
> (1) Firstly, the visualization results in the supplementary Fig.3 demonstrate that the adaptive reconstruction loss function effectively guides the model to learn fusion, **forming a complete supervisory signal** that encompasses scene information.  Models trained under the supervision of this proposed loss function can reconstruct efficient fused images, especially for obscured objects, such as individuals concealed by smoke.
>
> (2) **The analysis of the $\beta$** in the adaptive reconstruction function shows that setting it to $\text{Mean}(V)$ improves alignment with the visible domain. As shown in the following Table1, this is more effective than the learnable setting or $\text{Mean}(I)$, especially when combined with the style-adjustment fusion (SAF) module.
>
> Table 1: Ablation studies of $\beta$ in Eq.9 on RoadScene dataset.
> | Methods               | EN ↑  | SD ↑   | SF ↑   | Qbaf ↑ | VIF ↑  |
> |-----------------------|-------|--------|--------|--------|--------|
> | $\text{Mean}(I)$       | 7.31  | 53.72  | 17.61  | 0.52   | 0.70   |
> | Learnable             | 7.41  | 53.90  | 18.04  | 0.54   | 0.71   |
> | **$\text{Mean}(V)$ (Original)** | **7.55** | **55.29** | **18.32** | **0.56** | **0.72** |
>
> (3) We further analyze **the operation used to form the supervision signals** in the loss function.  As shown in the following Table2, Replacing the current max operation with a mean operation, or using separate supervisions for $R(I)$ and $R(V)$, both led to decreased performance. The key reason is that when using the mean or separate supervisions, the network tends to blend the two input signals too smoothly. This excessive blending results in a loss of important content details in the fused output. In contrast, the max operation better preserves the distinct content characteristics from the source modalities, preventing smoothing.
>
> Table 2: Ablation studies of operations in Eq.11 on RoadScene dataset.
> | Methods               | EN ↑  | SD ↑   | SF ↑   | Qbaf ↑ | VIF ↑  |
> |-----------------------|-------|--------|--------|--------|--------|
> | $\text{Mean}(\cdot)$           | 6.67  | 48.50  | 15.13  | 0.53   | 0.66   |
> | $\text{Separation}$            | 7.42  | 53.14  | 17.99  | 0.54   | 0.71   |
> | **Max(·) (Original)** | **7.55** | **55.29** | **18.32** | **0.56** | **0.72** |
>
> (4) Additionally, we study **the learnable $\alpha$**, which is utilized to adjust the entropy of source images.  An excessively large or small $\alpha$ damages the image priors of the supervision signal.  We conduct an additional experiment by removing the constraints on $\alpha$. The results show a significant performance decrease on RoadScene dataset, as shown in the following Table 3. This is due to excessive modulation of the source images, which cannot provide effective supervision, thereby demonstrating the necessity of the constraint.
>
> Table 3: Parameters $\alpha$  of adaptive reconstruction loss on RoadScene dataset.
> | $\alpha$         | EN    | SF   | Qbaf | VIF  | SSIM |
> |------------------|-------|------|------|------|------|
> | w/o constrain| 5.97  | 14.30 | 0.41 | 0.57 | 0.78 |
>  | **w/ constrain (Original)** | **7.55** | **18.32** | **0.56** | **0.72** | **1.21** |

---

> > ### Author Response · Authors · 2024-11-29
> > **Looking forward to your additional feedback**
> >
> > Dear Reviewer vKhE,
> >
> > Thank you once more for your insightful review.
> >
> > As the author-reviewer discussion period is nearing its end, we would greatly value your feedback on whether our revisions and responses have adequately addressed your earlier concerns.
> >
> > Should you have any further questions or additional feedback, please let us know, and we will address them promptly.
> >
> > We appreciate your time and contributions.
> >
> > Warm regards,
> >
> > The Authors

---

### Official Review · Reviewer_vmfZ · 2024-11-03

**Soundness:** 3
**Presentation:** 3
**Contribution:** 3
**Rating:** 5
**Confidence:** 4

**Summary:**

This paper proposes a novel style-coherent multi-modality fusion model, based on frequency analysis, which creatively decouples the style and content of modality for image fusion. This work argues that styles represent modality-specific differences in texture, saturation, and resolution, so the SNF module adaptively performs style preservation and enhancement during the fusion process, and SAF aligns cross-modal fused features to a designated modality, ensuring stylistic consistency. In addition, distinguishing from the traditional methods that directly supervise with source data, this work employs an adaptive reconstruction loss function.

**Strengths:**

1.The decoupling of style and content based on frequency domain analysis is creatively applied to image fusion, and the stylistic features of the source image are preserved and enhanced considering the characteristics of image fusion.

2. Adaptive reconstruction loss with good generalization is proposed.

**Weaknesses:**

1. This method has a preference for one modality in the fusion process, but how to choose the preference for two modalities whose dominance is not clear?

2. Poor interpretability of SAF modules and Losses.

**Questions:**

1. What’s advantageous of using information entropy-based clipping to constrain α?

2. What is the rationale for designing the fusion form in equation 8?

---

> ### Author Response · Authors · 2024-11-22
> **To Reviewer vmfZ**
>
> ``1. This method has a preference for one modality in the fusion process, but how to choose the preference for two modalities whose dominance is not clear? ``
>
> (1) SAF uses well-defined source distribution to **guide the different features into a unified domain** rather than the direct definition of the target domain. Fig. 7 in the main paper demonstrates that the fusion results do not completely depend on the aligned modality but instead more closely resemble its visual characteristics. It allows SAF to be more robust to variations in the source modalities.
>
> (2) The fused results demonstrate that SAF aligns images of different modalities while preserving complete scene information, with **differences in their visual characteristics.**  Fig. 4 in the supplementary shows that aligning with the infrared domain results in higher contrast, while aligning with the visible domain produces more visually satisfying images. This flexibility in visual characteristics further enhances the robustness of SAF to source modalities.
>
> (3) SAF allows for selecting the alignment domain **based on specific requirements**. Aligning with the domain containing critical information helps preserve essential original details and avoids excessive modulation. For example, when fusing visible and infrared images for human observation, it is preferable to align with the visible domain. This ensures that the visual information is well represented in the fused images (see Fig. 4 in the supplementary).
>
> `` 2. About equation 8. ``
>
> For SAF, Eq. 8 is used to adjust the distributional characteristics of the style-normalized fused feature to align with the visible feature $\hat{X}_V^L$. The learnable weight $W$ is utilized to channel-wise fuse $SN(X_V)$ and $SN(X_I)$, and then align the fused feature with the variance of the style-normalized $\hat{X}_V^L$. The mean of the style-normalized fused feature is adjusted to match the mean of $\hat{X}_I^L$. This process ensures that the fused output retains crucial information from modalities, optimized for the characteristics of the visible domain. We revise the main paper for more clarity.
>
> ``3. More analysis on SAF.``
>
> We conduct further analysis on SAF as follows.
>
> (1) For **feature alignment with a specific domain**, we present a visual comparison. Fig. 4 in the supplementary demonstrates that aligning modalities can reduce feature differences between modalities and does not lead to the loss of scene information.
>
> (2) For **the feature fusion process**, we conduct both visual and quantitative comparisons. As shown in Tab. 4 of the main paper, skipping the alignment and directly fusing results in significantly reduced performance, demonstrating the necessity of alignment for effective fusion. The comparison of visual results in supplementary Fig. 4 shows that fused features with alignment preserve complete scene information without over-smoothing.
>
> (3) For **SAF aligning different modalities**, we also conduct both visual and quantitative comparisons. The visualization results in supplementary Fig. 4 and the quantitative results in the following Table 1 highlight the advantages of selecting different modal domains for alignment. For IVF, although aligning with the visible domain provides more distinct details and contrast, maintaining excessively high contrast reduces visual performance. Therefore, prioritizing the information-abundant domain helps avoid the destruction of image priors caused by excessive modulation.
>
> Table 1: Ablation studies of the selection of alignment across different domains in SAF on RoadScene dataset.
>
> | SAF                           | EN ↑  | SF ↑   | Qbaf ↑ | VIF ↑  | SSIM ↑  |
> |-------------------------------|-------|--------|--------|--------|---------|
> | Align w/ infrared domain      | **7.63** | **19.06** | 0.53   | 0.68   | 1.03    |
> | Align w/ visible domain (Original) | 7.55  | 18.32   | **0.56** | **0.72** | **1.21** |

---

> ### Author Response · Authors · 2024-11-28
> **To Reviewer vmfZ**
>
> ``4. About proposed adaptive reconstruction loss.``
>
> Our proposed adaptive reconstruction loss function employs a learned linear rescaling function for the source input to form a unified and complete supervision signal. In Eq. 9, the learned linear rescaling function uses $\beta$ to align the center of the distribution of the visible image domain, while $\alpha$ aligns the variance of different images to adjust the entropy of the source images.
>
> The visualization results in the supplementary Fig.3 demonstrate that the rescaled source images effectively guides the model to learn fusion, **forming a complete supervisory signal** that encompasses scene information. Models trained under the supervision of this proposed loss function can reconstruct efficient fused images, especially for obscured objects, such as individuals concealed by smoke.
>
> We also conduct the analysis and ablation studies on the proposed loss and add them to the main paper and supplementary.
>
> (1) We analyze the **$\beta$ parameter** in the adaptive reconstruction function. Table 2 below shows that setting it to $\text{Mean}(V)$ improves alignment with the visible domain when combined with the style-adjustment fusion (SAF) module. This avoids aligned domain conflicts, making it more stable and effective than the learnable setting or $\text{Mean}(I)$.
>
> Table 2: Ablation studies of $\beta$ in Eq. 9 on RoadScene dataset.
>
> | Methods               | EN ↑  | SD ↑   | SF ↑   | Qbaf ↑ | VIF ↑  |
> |-----------------------|-------|--------|--------|--------|--------|
> | $\text{Mean}(I)$      | 7.31  | 53.72  | 17.61  | 0.52   | 0.70   |
> | Learnable             | 7.41  | 53.90  | 18.04  | 0.54   | 0.71   |
> | **$\text{Mean}(V)$ (Original)** | **7.55** | **55.29** | **18.32** | **0.56** | **0.72** |
>
> (2) In the following response, we further analyze the **$\alpha$ parameter**. Table 4 below demonstrates the importance of the constraint in maintaining image priors and forming effective implicit supervision.
>
> (3) We analyze **the operation** in the loss function. Table 3 below shows that the current $\text{Max}(\cdot)$ operation, compared to separate supervision and the mean operation, helps to obtain clearer supervision signals, thereby achieving promising results.
>
> Table 3: Ablation studies of operations in Eq. 11 on RoadScene dataset.
>
> | Methods | EN ↑ | SD ↑ | SF ↑ | Qbaf ↑ | VIF ↑ |
> |-------------------------|-------|--------|--------|--------|--------|
> | $\text{Mean}(\cdot)$ | 6.67 | 48.50 | 15.13 | 0.53 | 0.66 |
> | $\text{Separation}$ | 7.42 | 53.14 | 17.99 | 0.54 | 0.71 |
> | **$\text{Max}(\cdot)$  (Original)** | **7.55** | **55.29** | **18.32** | **0.56** | **0.72** |
>
> ``5. Advantages of using information entropy-based clipping to constrain α.``
>
> The learnable parameter $\alpha$ is utilized to adjust the entropy of the image. An excessively large or small $\alpha$ damages the image priors of the supervision signal. Variance is an important indicator of the amount of information entropy. Such constraints make the source image with smaller variances effectively align with other source images. It ensures that the adjusted information entropy of supervision signals tends toward consistency, thereby ensuring the integrity of the supervision signal. We conduct an additional experiment by removing the constraints on $\alpha$. The results show a significant performance decrease on RoadScene dataset, as shown in the following Table 4. This is due to excessive modulation of the source images, which cannot provide effective supervision, thereby demonstrating the necessity of the constraint.
>
> Table 4: Parameters $\alpha$ of adaptive reconstruction loss on RoadScene dataset.
>  | $\alpha$ | EN ↑ | SF ↑ | Qbaf ↑ | VIF ↑ | SSIM ↑ |
> |-------------------------|-------|--------|--------|--------|--------|
> | w/o constrain | 5.97 | 14.30 | 0.41 | 0.57 | 0.78 |
> | w/ constrain (Original) | **7.55** | **18.32** | **0.56** | **0.72** | **1.21** |

---

> > ### Author Response · Authors · 2024-11-29
> > **Looking forward to your additional feedback**
> >
> > Dear Reviewer vmfZ,
> >
> > Thank you once more for your insightful review.
> >
> > As the author-reviewer discussion period is nearing its end, we would greatly value your feedback on whether our revisions and responses have adequately addressed your earlier concerns.
> >
> > Should you have any further questions or additional feedback, please let us know, and we will address them promptly.
> >
> > We appreciate your time and contributions.
> >
> > Warm regards,
> >
> > The Authors

---

### Author Response · Authors · 2024-11-22
**To reviewers and area chairs**

We thank all reviewers and area chairs for their valuable time. We are glad to find that reviewers recognized the following merits of our work:
+  **Innovative contribution and strong motivation [vmfZ, vKhE, ooZi]**: Our proposed SCFNet addresses the challenges of MMIF by effectively demonstrating the importance of coherent styles and integrating complete content.

   - This paper proposes a novel style-coherent multi-modality fusion model, based on frequency analysis, which creatively decouples the style and content of modality for image fusion. [**vmfZ**]

   - The style-coherent approach is applied to multimodal fusion field and valid to be effective.[**vKhE**]

   - The proposed approach of separating style and content is sound and promising.[**ooZi**]


+   **Impressive performance [ vKhE, Exws, ooZi]**: Experimental results indicate that SCF surpasses existing MMIF methods in various scenes and tasks.
    - Experimental results show that this method outperforms existing approaches, indicating strong potential for various image processing applications.[**vmfZ**]
    - The performance of this paper seems better compared to some related SOTA works.[**Exws**]
    -  The results on several benchmarks suggest significant improvements over the state of the art.[**ooZi**]

+   **Well-written[vmfZ]**: The paper is well-written and easy to follow.

We sincerely thank all reviewers for their constructive feedback. Alongside the detailed point-by-point responses provided below, we have summarized the key revisions made in the rebuttal based on the reviewers' suggestions:

+ We added details of the **adaptive reconstruction loss** and further analyzed it with more ablation studies in the supplementary.[vmfZ,vKhE]
+ We provided analyses and experiments to clarify **style-alignment module**, including visual results and ablation studies in the supplementary. [vmfZ]
+ We included a summary of related **decomposition methods** and highlighted the differences from our method.[Exws, ooZi]
+ We explained the **Fourier prior embedded block** and included a related description in the main paper.[ooZi]
+ We incorporate visual results of fused features in the main paper to emphasize the motivation and contribution.[ooZi]
+ We elaborated on the details of the implementation of our experimental results.[Exws]
+ We corrected ambiguous expressions and typos in the main text.[Exws]

Again, we thank all Reviewers and Area Chairs!

Best regards,

Authors

---

### Author Response · Authors · 2024-12-02
**To Reviewers**

Dear Reviewers,

Thank you once again for your feedback on our submission. We hope our responses have addressed the concerns you raised. As today marks the final day of the discussion period, we would like to kindly invite any additional questions or suggestions you may have.

We greatly appreciate your time, advice, and support.

Best regards,

Authors

---

### Meta-Review · Area_Chair_WQUQ · 2024-12-17

**Metareview:**

This paper aims to address the issue of style discrepancies and proposes a method to construct style-coherent multi-modality fusion model via frequency analysis. The proposed model utilizes a dual-branch encoder architecture, incorporating FPE, SNF and SAF modules to enhance content representation and align features across modalities. Experiments on different tasks are performed to evaluate the performance of the proposed method.

Reviewers expressed their interest in applying style-coherent approach to multimodal fusion. However, all reviewers raised concerns about the rationale and interpretation of the designed key modules (e.g., SAF), the fusion operations (e.g., Equation 8) and specific losses (e.g., adaptive reconstruction loss). According to the reviewers’ comments, sufficient analysis and comprehensive evaluation are required to further explain and illustrate the rationale and logic of the proposed method, and these explanations and illustrations need to be clearly presented or pointed out in the main text. Based on the above considerations, I think the current manuscript does not match the ICLR’s requirement and I do not recommend to accept this manuscript.

**Additional Comments On Reviewer Discussion:**

Two reviewers gave marginally positive ratings, one of which increased the rating to 6 during the rebuttal period, but two reviewers gave negative ratings. All reviewers raised concerns about the rationale and interpretation of the designed modules, the processes and losses. Although authors provide responses, sufficient analysis and comprehensive evaluations still need to be clearly presented or pointed out in the main text.

---

### Decision · Program_Chairs · 2025-01-22

Reject